# Identifiable Latent Causal Content for Domain Adaptation under Latent Covariate Shift

## Abstract

Multi-source domain adaptation (MSDA) addresses the challenge of learning a label prediction function for an unlabeled target domain by leveraging both the labeled data from multiple source domains and the unlabeled data from the target domain. Conventional MSDA approaches often rely on covariate shift or conditional shift paradigms, which assume a consistent label distribution across domains. However, this assumption proves limiting in practical scenarios where label distributions do vary across domains, diminishing its applicability in real-world settings. For example, animals from different regions exhibit diverse characteristics due to varying diets and genetics.

Motivated by this, we propose a novel paradigm called latent covariate shift (LCS), which introduces significantly greater variability and adaptability across domains. Notably, it provides a theoretical assurance for recovering the latent cause of the label variable, which we refer to as the latent content variable. Within this new paradigm, we present an intricate causal generative model by introducing latent noises across domains, along with a latent content variable and a latent style variable to achieve more nuanced rendering of observational data. We demonstrate that the latent content variable can be identified up to block identifiability due to its versatile yet distinct causal structure. We anchor our theoretical insights into a novel MSDA method, which learns the label distribution conditioned on the identifiable latent content variable, thereby accommodating more substantial distribution shifts. The proposed approach showcases exceptional performance and efficacy on both simulated and real-world datasets.

## 1 Introduction

Multi-source domain adaptation (MSDA) aims to utilize labeled data from multiple source domains and unlabeled data from the target domain, to learn a model to predict well in the target domain. Formally, denoting the input as $\mathbf{x}$ (e.g., an image), $\mathbf{y}$ as labels in both source and target domains, and $\mathbf{u}$ as the domain index, during MSDA training, we have labeled source domain input-output pairs, $(\mathbf{x}^{\mathcal{S}}, \mathbf{y}^{\mathcal{S}})$, drawn from source domain distributions $p_{\mathbf{u}=\mathbf{u}1}(\mathbf{x}, \mathbf{y}), ..., p_{\mathbf{u}=\mathbf{u}_m}(\mathbf{x}, \mathbf{y}), ...$[1] Note that the distribution $p_{\mathbf{u}}(\mathbf{x}, \mathbf{y})$ may vary across domains. Additionally, we observe some unlabeled target domain input data, $\mathbf{x}^{\mathcal{T}}$, sampled from the target domain distribution $p_{\mathbf{u}\mathcal{T}}(\mathbf{x}, \mathbf{y})$.

The success of MSDA hinges on two crucial elements: *variability* in the distribution $p_{\mathbf{u}}(\mathbf{x}, \mathbf{y})$, determining the extent to which it may differ across domains, and the imperative of *invariability* in a certain portion of the same distribution to ensure effective adaptation to the target domain. Neglecting to robustly capture this invariability, often necessitating a theoretical guarantee, hampers adaptability and performance in the target domain. Our approach comprehensively addresses both these elements, which will be elaborated upon in subsequent sections.

MSDA can be broadly classified into two primary strands: *Covariate Shift* (Huang et al., 2006; Bickel et al., 2007; Sugiyama et al., 2007; Wen et al., 2014) and *Conditional Shift* (Zhang et al., 2013; 2015; Schölkopf et al., 2012; Stojanov et al., 2021; Peng et al., 2019). In the early stages of MSDA research, MSDA methods focus on the first research strand *Covariate Shift* as depicted

---

[1]To simplify our notation without introducing unnecessary complexity, we employ the notation $p_{\mathbf{u}=\mathbf{u}_m}(\mathbf{x}, \mathbf{y})$ to denote $p(\mathbf{x}, \mathbf{y}|\mathbf{u} = \mathbf{u}_m)$, and express $\mathbf{u} = \mathbf{u}_m$ as $\mathbf{U} = \mathbf{u}_m$ with the aim of clarity.

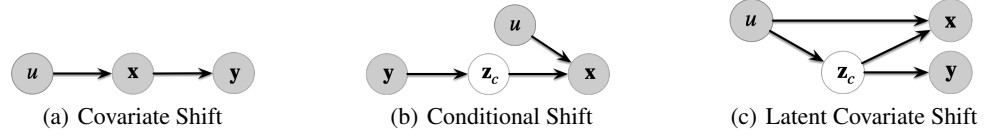

|  | | |
|---|---|---|
| (a) Covariate Shift | (b) Conditional Shift | (c) Latent Covariate Shift |

Figure 1: The illustration of three different paradigms for MSDA. Covariate Shift: $p_\mathbf{u}(\mathbf{x})$ changes across domains, while $p_\mathbf{u}(\mathbf{y}|\mathbf{x})$ is invariant across domains. Conditional Shift: $p_\mathbf{u}(\mathbf{y})$ is invariant, while $p_\mathbf{u}(\mathbf{x}|\mathbf{y})$ changes across domains. Latent Covariate Shift: $p_\mathbf{u}(\mathbf{z}_c)$ changes across domains while $p_\mathbf{u}(\mathbf{y}|\mathbf{z}_c)$ is invariant.

by Figure 1(a). It assumes that $p_\mathbf{u}(\mathbf{x})$ changes across domains, while the conditional distribution $p_\mathbf{u}(\mathbf{y}|\mathbf{x})$ remains invariant across domains. However, this assumption does not always hold in practical applications, such as image classification. For example, the assumption of invariant $p_\mathbf{u}(\mathbf{y}|\mathbf{x})$ implies that $p_\mathbf{u}(\mathbf{y})$ should change as $p_\mathbf{u}(\mathbf{x})$ changes. Yet, we can easily manipulate style information (e.g., hue, view) in images to alter $p_\mathbf{u}(\mathbf{x})$ while keeping $p_\mathbf{u}(\mathbf{y})$ unchanged, which clearly contradicts the assumption. In contrast, most recent works delve into the *Conditional Shift* as depicted by Figure 1(b). It assumes that the conditional $p_\mathbf{u}(\mathbf{x}|\mathbf{y})$ changes while $p_\mathbf{u}(\mathbf{y})$ remains invariant across domains (Zhang et al., 2013; 2015; Schölkopf et al., 2012; Stojanov et al., 2021; Peng et al., 2019). Consequently, it has spurred a popular class of methods focused on learning invariant representations across domains to target the latent content variable $\mathbf{z}_c$ in Figure 1(b) (Ganin et al., 2016; Zhao et al., 2018; Saito et al., 2018; Mancini et al., 2018; Yang et al., 2020; Wang et al., 2020; Li et al., 2021; Kong et al., 2022). However, the label distribution $p_\mathbf{u}(\mathbf{y})$ can undergo changes across domains in many real-world scenarios (Tachet des Combes et al., 2020; Lipton et al., 2018; Zhang et al., 2013), and enforcing invariant representations can degenerate the performance (Zhao et al., 2019).

In many real-world applications, the label distribution $p_\mathbf{u}(\mathbf{y})$ exhibits variation across different domains. For instance, diverse geographical locations entail distinct species sets and/or distributions. This characteristic is well-illustrated by a recent, meticulously annotated dataset focused on studying visual generalization across various locations (Beery et al., 2018). Their findings illuminate that the species distribution presents a long-tailed pattern at each location, with each locale featuring a distinct and unique distribution [2]. Label distribution shifts can also be corroborated through analysis on the WILDS benchmark dataset, which investigates shifts in distributions within untamed and unregulated environments (Koh et al., 2021). This study uncovers pronounced disparities in label distributions between Africa and other regions [3]. These distinctions encompass a notable decrease in recreational facilities and a marked rise in single-unit residential properties.

To enhance both critical elements of MSDA, namely variability and invariability, we introduce a novel paradigm termed *Latent Covariate Shift (LCS)*, as depicted in Figure 1(c). Unlike previous paradigms, LCS introduces a latent content variable $\mathbf{z}_c$ as the common cause of $\mathbf{x}$ and $\mathbf{y}$. The distinction between LCS and previous paradigms is detailed in Section 2. In essence, LCS allows for the flexibility for $p_\mathbf{u}(\mathbf{z}_c)$, $p_\mathbf{u}(\mathbf{x})$, $p_\mathbf{u}(\mathbf{y})$ and $p_\mathbf{u}(\mathbf{x}|\mathbf{z}_c)$ to vary across domains (greater variability). Simultaneously, its inherent causal structure guarantees that $p_\mathbf{u}(\mathbf{y}|\mathbf{z}_c)$ remains invariant irrespective of the domain (invariability with assurance). This allowance for distributional shifts imparts versatility and applicability to a wide array of real-world problems, while the stability of $p_\mathbf{u}(\mathbf{y}|\mathbf{z}_c)$ stands as the pivotal factor contributing to exceptional performance across domains.

Within this new paradigm, we present an intricate latent causal generative model, by introducing the latent style variable $\mathbf{z}_s$ in conjunction with $\mathbf{z}_c$, as illustrated in Figure 2(a). We delve into an extensive analysis of the identifiability within our proposed causal model, affirming that the latent content variable $\mathbf{z}_c$ can be established up to block identifiability[4] through rigorous theoretical examination. Since $\mathbf{z}_s$ and the label become independent given $\mathbf{z}_c$, it is no need to recover $\mathbf{z}_c$. The identifiability on $\mathbf{z}_c$ provides a solid foundation for algorithmic designs with robust theoretical assurances. Subsequently, we translate these findings into a novel method that learns an invariant conditional distribution $p_\mathbf{u}(\mathbf{y}|\mathbf{z}_c)$, known as independent causal mechanism, for MSDA. Leveraging the guaranteed identifiability of $\mathbf{z}_c$, our proposed method ensures principled generalization to the target domain. Empirical evaluation on both synthetic and real-world data showcases the effectiveness of our approach, outperforming state-of-the-art methods.

---

[2] For a comprehensive depiction of these distributions, please refer to Figure 4 in Beery et al. (2018). Additionally, the APPENDIX offers an analogous representation, as shown in Figure 6 for completeness.

[3] For a thorough comparison of distributions, refer to Figures 24 and 27 in Koh et al. (2021).

[4] There exists an invertible function between the recovered $\mathbf{z}_c$ and the true one (Von Kügelgen et al., 2021).

## 2 RELATED WORK

**Learning invariant representations**. Due to the limitations of covariate shift, particularly in the context of image data, most current research on domain adaptation primarily revolves around addressing conditional shift. This approach focuses on learning invariant representations across domains, a concept explored in works such as Ganin et al. (2016); Zhao et al. (2018); Saito et al. (2018); Mancini et al. (2018); Yang et al. (2020); Wang et al. (2020); Li et al. (2021); Wang et al. (2022b); Zhao et al. (2021). These invariant representations are typically obtained by applying appropriate linear or nonlinear transformations to the input data. The central challenge in these methods lies in enforcing the invariance of the learned representations. Various techniques are employed to achieve this, such as maximum classifier discrepancy (Saito et al., 2018), domain discriminator for adversarial training (Ganin et al., 2016; Zhao et al., 2018; 2021), moment matching (Peng et al., 2019), and relation alignment loss (Wang et al., 2020). However, all these methods assume label distribution invariance across domains. Consequently, when label distributions vary across domains, these methods may perform well only in the overlapping regions of label distributions across different domains, encountering challenges in areas where distributions do not overlap. To overcome this, recent progress focuses on learning invariant representations conditional on the label across domains (Gong et al., 2016; Ghifary et al., 2016; Tachet des Combes et al., 2020). One of the challenges in these methods is that the labels in the target domain are unavailable. Moreover, these methods do not guarantee that the learned representations align consistently with the true relevant information.

**Learning invariant conditional distribution** $p_{\mathbf{u}}(\mathbf{y}|\mathbf{z}_c)$. The investigation of learning invariant conditional distributions, specifically $p_{\mathbf{u}}(\mathbf{y}|\mathbf{z}_c)$, for domain adaptation has seen limited attention compared to the extensive emphasis on learning invariant representations (Kull & Flach, 2014; Bouvier et al., 2019). What sets our proposed method apart from these two works is its causal approach, providing identifiability for the true latent content $\mathbf{z}_c$. This serves as a theoretical guarantee for capturing invariability, addressing the second key element of MSDA. It ensures that the learned $p_{\mathbf{u}}(\mathbf{y}|\mathbf{z}_c)$ in our work can generalize to the target domain in a principled manner. In addition, certain studies explore the identification of $p_{\mathbf{u}}(\mathbf{y}|\mathbf{z}_c)$ by employing a proxy variable, as demonstrated by (Alabdulmohsin et al., 2023). The challenge of these studies lies in devising an efficient proxy variable. In contrast, although the work do not need such proxy variable, it is worth noting that our work may necessitate stronger assumptions for identifying latent $\mathbf{z}_c$, compared with proxy based methods. Additionally, our Latent Causal Structure (LCS) allows for more flexibility in accommodating variability, addressing the first key element of MSDA. Besides, in the context of out-of-distribution generalization, some recent works have explored the learning of invariant conditional distributions $p_{\mathbf{u}}(\mathbf{y}|\mathbf{z}_c)$ (Arjovsky et al., 2019; Sun et al., 2021; Liu et al., 2021; Lu et al., 2021). For example, Arjovsky et al. (2019) impose learning the optimal invariant predictor across domains, while the proposed method directly explores conditional invariance by the proposed latent causal model. Moreover, some recent works design specific causal models tailored to different application scenarios from a causal perspective. For example, Liu et al. (2021) mainly focus on a single source domain, while the proposed method considers multiple source domains. The work by Sun et al. (2021) explores the scenarios where a confounder to model the causal relationship between latent content variables and style variables, while the proposed method considers the scenarios in which latent style variable is caused by content variable. The work in (Lu et al., 2021) focuses on the setting where label variable is treated as a variable causing the other latent variables, while in our scenarios label variable has no child nodes.

**Causality for Domain Generalization** A strong connection between causality and generalization has been established in the literature (Peters et al., 2016). Building upon this insight, current research has leveraged causality to introduce novel methods across various applications, including domain generalization (Mahajan et al., 2021; Christiansen et al., 2021; Wang et al., 2022a), text classification (Veitch et al., 2021), and Out-of-Distribution Generalization (Ahuja et al., 2021). Among these applications, domain generalization is closely related to our problem setting. However, it involves scenarios where the input data in the target domain cannot be directly observed. The lack of access to the input data makes obtaining identifiability results challenging, and is left for future work.

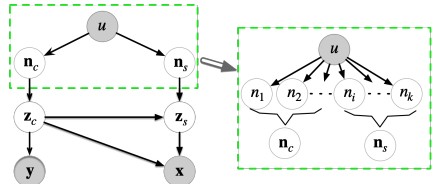 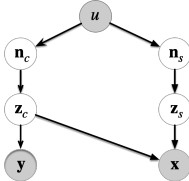

(a) The proposed causal model  (b) An equivalent graph structure

Figure 2: (a) The proposed latent causal model, which splits latent noise variables $\mathbf{n}$ into two disjoint parts, $\mathbf{n}_c$ and $\mathbf{n}_s$. (b) An equivalent graph structure, which can generate the same observed data $\mathbf{x}$ as obtained by (a), resulting in a non-identifiability result.

# 3 THE PROPOSED LATENT CAUSAL MODEL FOR LCS

LCS is a new paradigm in MSDA, enriching the field with elevated variability and versatility. Within this innovative framework, we're presented with an opportunity to delve into more intricate models. This section is dedicated to presenting a refined causal model, tailored to this paradigm.

Drawing inspiration from the world of artistic representation, we integrate a latent style variable, working in tandem with the latent content variable. Consider the profound contrast between the drawing styles employed in a close-up portrayal of a human figure and those used to depict a vast, distant mountain vista. This stark divergence vividly underscores how the underlying content intricately molds the expressive nuances in artistic representation.

Fundamentally, the latent content exerts direct influence over the available styles. To account for this, we introduce a latent style variable, seamlessly integrated as a direct descendant of the latent content variable, as depicted in Figure 2(a). Concurrently, the observed domain variable $\mathbf{u}$ serves as an indicator of the specific domain from which the data is sourced. This domain variable gives rise to two distinct groups of latent noise variables: latent content noise, denoted as $\mathbf{n}_c$, and latent style noise, denoted as $\mathbf{n}_s$. Analogous to exogenous variables in causal systems, these elements play pivotal roles in shaping both the latent content variable $\mathbf{z}_c$ and the latent style variable $\mathbf{z}_s$.

This model stands out in terms of versatility and applicability across a wide range of real-world scenarios, surpassing the capabilities of models in previous paradigms including covariate shift and conditional shift. This is because it accommodates variations in $p_{\mathbf{u}}(\mathbf{z}_c)$, $p_{\mathbf{u}}(\mathbf{x})$ and $p_{\mathbf{u}}(\mathbf{x}|\mathbf{z}_c)$ across domains. Moreover, it provides a theoretically established guarantee of invariability for $p_{\mathbf{u}}(\mathbf{y}|\mathbf{z}_c)$ independent of the domain (see Sections 4 and 5). This pivotal property facilitates the model's ability to generalize predictions across a diverse array of domains. To parameterize it, we make the assumption that $\mathbf{n}$ follows an exponential family distribution given $\mathbf{u}$, and we describe the generation of $\mathbf{z}$ and $\mathbf{x}$ as follows:

$$p_{(\mathbf{T},\boldsymbol{\eta})}(\mathbf{n}|\mathbf{u}) = \prod_{i=1}^{\ell} \frac{1}{Z_i(\mathbf{u})} \exp[\sum_{j=1}^{2} (T_{i,j}(n_i)\eta_{i,j}(\mathbf{u}))], \tag{1}$$

$$\mathbf{z}_c = \mathbf{g}_c(\mathbf{n}_c), \ \mathbf{z}_s = \mathbf{g}_{s2}(\mathbf{g}_{s1}(\mathbf{z}_c) + \mathbf{n}_s), \tag{2}$$

$$\mathbf{x} = \mathbf{f}(\mathbf{z}_c, \mathbf{z}_s) + \boldsymbol{\varepsilon}. \tag{3}$$

In Eq. 1, $Z_i(\mathbf{u})$ represents the normalizing constant, $T_{i,j}(n_i)$ stands as the sufficient statistic for $n_i$, and $\ell$ denotes the number of latent noises. The natural parameter $\eta_{i,j}(\mathbf{u})$ is dependent on the domain variable $\mathbf{u}$. To establish a coherent connection between these latent noise variables $\mathbf{n}$ and the latent variables $\mathbf{z}$, we employ post-nonlinear models, as defined in Eq. 2, where $\mathbf{g}_c$ and $\mathbf{g}_{s2}$ are invertible functions. This concept of post-nonlinear models (Zhang & Hyvarinen, 2012) represents a generalized form of additive noise models (Hoyer et al., 2008), which find widespread use across various domains. Furthermore, our proposed model incorporates two fundamental causal assumptions that serve to underscore its novelty and distinctiveness within the field of latent causal modeling.

$\mathbf{z}_c$ **causes** $\mathbf{y}$: Prior research has often considered the causal relationship between $\mathbf{x}$ and $\mathbf{y}$ as $\mathbf{y} \rightarrow \mathbf{x}$ (Gong et al., 2016; Stojanov et al., 2019; Li et al., 2018). In contrast, our approach employs $\mathbf{z}_c \rightarrow \mathbf{y}$. Note that these two cases are not contradictory, as they pertain to different interpretations of the

labels $\mathbf{y}$ representing distinct physical meanings. To clarify this distinction, let us denote the label in the first case as $\hat{\mathbf{y}}$ (i.e., $\hat{\mathbf{y}} \rightarrow \mathbf{x}$) to distinguish it from $\mathbf{y}$ in the second case (i.e., $\mathbf{z}_c \rightarrow \mathbf{y}$). In the first case, consider the generative process for images: a label, $\hat{\mathbf{y}}$, is initially sampled, followed by the determination of content information based on this label, and finally, the generation of an image. This sequence aligns with reasonable assumptions in real-world applications. In our proposed latent causal model, we introduce $\mathbf{n}_c$ to play a role similar to $\hat{\mathbf{y}}$ and establish a causal connection with the content variable $\mathbf{z}_c$. In the second case, $\mathbf{z}_c \rightarrow \mathbf{y}$ represents the process where experts extract content information from provided images and subsequently assign appropriate labels based on their domain knowledge. This assumption has been adopted by recent works (Mahajan et al., 2021; Liu et al., 2021; Sun et al., 2021). Notably, both of these distinct labels, $\hat{\mathbf{y}}$ and $\mathbf{y}$, have been concurrently considered in Mahajan et al. (2021). In summary, these two causal models, $\hat{\mathbf{y}} \rightarrow \mathbf{x}$ and $\mathbf{z}_c \rightarrow \mathbf{y}$, capture different aspects of the generative process and are not inherently contradictory, reflecting varying perspectives on the relationship between labels and data generation.

$\mathbf{z}_c$ **causes** $\mathbf{z}_s$**:** As discussed earlier, the underlying content directly molds the styles. Therefore, we adopt the causal relationship where $\mathbf{z}_c$ serves as the cause of $\mathbf{z}_s$. This can be interpreted as the essence of the object, $\mathbf{z}_c$, being the primary factor from which a latent style variable, $\mathbf{z}_s$, emerges to ultimately render the observation $\mathbf{x}$. This is aligned with previous works in domain adaptation field (Gong et al., 2016; Stojanov et al., 2019; Mahajan et al., 2021), as well as with recent advancements in self-supervised learning (Von Kügelgen et al., 2021; Daunhawer et al., 2023).

## 4 IDENTIFIABILITY ANALYSIS

In this section, we provide an identifiability analysis for the proposed latent causal model in section 3. We commence by illustrating that attaining complete identifiability for the proposed causal model proves unattainable without the imposition of stronger assumptions. This assertion is substantiated by the construction of an alternative solution that deviates from the true latent causal variables, yet yields the same observational data. Following this, we move forward to affirm that achieving partial identifiability of the latent content variables, denoted as $\mathbf{z}_c$, up to block identifiability is within reach. This level of identifiability already provides a solid foundation for guiding algorithm designs with robust theoretical guarantees, while maintaining the necessary flexibility for adaptation. Indeed, we can see that $\mathbf{z}_c$ serves as a common cause for both $\mathbf{y}$ and $\mathbf{z}_s$, illustrating a typical scenario of spurious correlation between $\mathbf{y}$ and $\mathbf{z}_s$. Once $\mathbf{z}_c$ is successfully recovered, it alone contains ample information to make accurate predictions about the label. Given $\mathbf{z}_c$, the label and the latent style become independent. Consequently, it is unnecessary and potentially counterproductive to recover the latent style variable for label prediction, when the latent content has already been retrieved.

### 4.1 COMPLETE IDENTIFIABILITY: THE NON-IDENTIFIABILITY RESULT

Given the proposed causal model, one of the fundamental problems is whether we can uniquely recover the latent variables, i.e., identifiability. We show that achieving complete identifiability of the proposed latent causal model remains a challenging endeavor, as follows:

**Proposition 4.1** *Suppose that the latent causal variables $\mathbf{z}$ and the observed variable $x$ follow the latent causal models defined in Eq. 1-3, given observational data distribution $p(\mathbf{y}, \mathbf{x}|\mathbf{u})$, there exists an alternative solution, which can yield exactly the same observational distribution, resulting in non-identifiablity, without further assumptions.*

**Proof sketch**  The proof of proposition 4.1 can be done by proving that we can always construct another alternative solution to generate the same observation, leading to the non-identifiability result. The alternative solution can be constructed by removing the edge from $\mathbf{z}_c$ to $\mathbf{z}_s$ as depicted by Figure 2(b), i.e., $\mathbf{z}'_c = \mathbf{z}_c = \mathbf{g}_c(\mathbf{n}_c)$, $\mathbf{z}'_s = \mathbf{n}_s$, and the mixing mapping from $\mathbf{z}'$ to $\mathbf{x}$ is a composition function, e.g., $\mathbf{f} \circ \mathbf{f}'$ where $\mathbf{f}'(\mathbf{z}') = [\mathbf{z}'_c, \mathbf{g}_{s2}(\mathbf{g}_{s1}(\mathbf{z}'_c) + \mathbf{z}'_s)]$.

**Intuition**  The non-identifiability result above is because we can not determine which path is the correct path corresponding to the net effect of $\mathbf{n}_c$ on $\mathbf{x}$; *i.e..*, both $\mathbf{n}_c \rightarrow \mathbf{z}_c \rightarrow \mathbf{z}_s \rightarrow \mathbf{x}$ in Figure 2(a) and $\mathbf{n}_c \rightarrow \mathbf{z}_c \rightarrow \mathbf{x}$ in Figure 2(b) can generate the same observational data $\mathbf{x}$. This problem often appears in latent causal discovery and seriously hinders the identifiability of latent causal models.

## 4.2 PARTIAL IDENTIFIABILITY: IDENTIFYING $\mathbf{z}_c$

While the above result in proposition 4.1 shows that achieving complete identifiability may pose challenges, for the specific context of domain adaptation, our primary interest lies in the identifiability of $\mathbf{z}_c$, rather than the latent style variable $\mathbf{z}_s$. This focus is justified by the fact that the label $\mathbf{y}$ is solely caused by $\mathbf{z}_c$. In fact, we have established the following partial identifiability result:

**Proposition 4.2** *Suppose latent causal variables $\mathbf{z}$ and the observed variable $\mathbf{x}$ follow the generative models defined in Eq. 1- Eq. 3. Assume the following holds:*

   *(i) The set $\{\mathbf{x} \in \mathcal{X} | \varphi_\varepsilon(\mathbf{x}) = 0\}$ has measure zero (i.e., has at the most countable number of elements), where $\varphi_\varepsilon$ is the characteristic function of the density $p_\varepsilon$.*

   *(ii) The function $\mathbf{f}$ in Eq. 3 is bijective.*

   *(iii) There exist $2\ell + 1$ distinct points $\mathbf{u}_0, \mathbf{u}_1, ..., \mathbf{u}_{2\ell}$ such that the matrix*
$$\mathbf{L} = (\boldsymbol{\eta}(\mathbf{u} = \mathbf{u}_1) - \boldsymbol{\eta}(\mathbf{u} = \mathbf{u}_0), ..., \boldsymbol{\eta}(\mathbf{u} = \mathbf{u}_{2\ell}) - \boldsymbol{\eta}(\mathbf{u} = \mathbf{u}_0)) \qquad (4)$$
   *of size $2\ell \times 2\ell$ is invertible where $\boldsymbol{\eta}(\mathbf{u}) = [\eta_{i,j}(\mathbf{u})]_{i,j}$,*

*then the recovered latent content variables $\hat{\mathbf{z}}_c$, which are learned by matching the true marginal data distribution $p(\mathbf{x}|\mathbf{u})$ and by using the dependence between $\mathbf{n}_c^{\mathcal{S}}$ and $\mathbf{y}^{\mathcal{S}}$ conditional on $\mathbf{u}$, are related to the true latent causal variables $\mathbf{z}_c$ by the following relationship: $\mathbf{z}_c = \mathbf{h}(\hat{\mathbf{z}}_c)$, where $\mathbf{h}$ denotes an invertible mapping.*

**Remark 4.3** *With access to label $\mathbf{y}$ in source domains, and the identified $\mathbf{z}_c$ in those domains, it becomes possible to accurately estimate the parameters of the conditional distribution $p(\mathbf{y}|\mathbf{z}_c)$. This allows for a robust modeling of the relationship between the latent content variables $\mathbf{z}_c$ and the corresponding labels $\mathbf{y}$ in the source domains. Then since $\mathbf{z}_c$ in target domain is also identified, and leveraging the invariance of $p(\mathbf{y}|\mathbf{z}_c)$, the learned conditional distribution $p(\mathbf{y}|\mathbf{z}_c)$ theoretically be generalized to the target domain. This generalization is grounded in the notion that the causal relationship between $\mathbf{z}_c$ and $\mathbf{y}$ remains consistent across domains.*

**Proof sketch** The proof of Proposition 4.2 can be outlined as follows: Given that the mapping from $\mathbf{n}$ to $\mathbf{z}$ is invertible, and in consideration of assumptions (i)-(iii), we can leverage results from nonlinear ICA (Hyvarinen et al., 2019; Khemakhem et al., 2020; Sorrenson et al., 2020). This implies that $\mathbf{n}$ can be identified up to permutation and scaling, i.e., $\mathbf{n} = \mathbf{P}\hat{\mathbf{n}} + \mathbf{c}$, where $\hat{\mathbf{n}}$ represents the recovered latent noise variables. Furthermore, the graph structure depicted in Figure 2(a) implies that $\mathbf{y}^{\mathcal{S}}$ is dependent on $\mathbf{n}_c^{\mathcal{S}}$ but independent of $\mathbf{n}_s^{\mathcal{S}}$, given $\mathbf{u}$. This insight enables us to eliminate the permutation indeterminacy, thus allowing us to identify $\mathbf{n}_c$. Ultimately, since the mapping from $\mathbf{n}_c$ to $\mathbf{z}_c$ is invertible as defined in Eq. 2, we can conclude the proof.

**Intuition** We note that all three assumptions align with standard practices in the nonlinear ICA literature (Hyvarinen et al., 2019; Khemakhem et al., 2020). This alignment is possible because nonlinear ICA and latent causal models naturally converge through the independence of components in the nonlinear ICA domain and the independence among exogenous variables in causal systems. However, there exists a crucial distinction: while nonlinear ICA seeks to identify *independent* latent variables, our objective in this context is to pinpoint the latent content variable within a causal system that *allows for causal relationships among latent variables*. This inherently renders the task more challenging. Fortunately, this discrepancy can be reconciled through two pivotal conditions: 1) the mapping from $\mathbf{n}$ to $\mathbf{z}$ is made invertible by constraining the function class, as defined in Eq. 2. 2) The conditional independence between $\mathbf{n}_s^{\mathcal{S}}$ and $\mathbf{y}^{\mathcal{S}}$, given $\mathbf{u}$, as depicted in the graph structure shown in Figure 2(a). This condition effectively eliminates the permutation indeterminacy typically encountered in nonlinear ICA. Furthermore, Proposition 4.2 establishes the existence of an invertible transformation between the recovered $\hat{\mathbf{z}}_c$ and the true latent content variable $\mathbf{z}_c$. Importantly, this invertible transformation has no bearing on domain adaptation tasks, as it indicates that $\hat{\mathbf{z}}_c$ encapsulates all and only the information pertaining to $\mathbf{z}_c$.

## 5 LEARNING INDEPENDENT CAUSAL MECHANISM $p_\mathbf{u}(\mathbf{y}|\mathbf{z}_c)$ FOR MSDA

The identifiability of $\mathbf{z}_c$ provides a principled foundation to ensure that we can effectively learn the independent causal mechanism $p_\mathbf{u}(\mathbf{y}|\mathbf{z}_c)$ across diverse domains. This, in turn, facilitates robust

generalization to the target domain. Additionally, given the invertible mapping between $\mathbf{n}_c$ and $\mathbf{z}_c$ as outlined in Eq. 2, the task of learning $p_{\mathbf{u}}(\mathbf{y}|\mathbf{z}_c)$ can be equivalently reformulated as the task of learning $p_{\mathbf{u}}(\mathbf{y}|\mathbf{n}_c)$. This alternative formulation remains consistent and invariant across various domains, as illustrated in Figure 2(a).

As elucidated in Proposition 4.2, the learned latent content variables $\hat{\mathbf{z}}_c$ are obtained by aligning with the true marginal data distribution $p(\mathbf{x}|\mathbf{u})$. Consequently, it logically follows that the acquired latent noise variables $\hat{\mathbf{n}}_c$ should also be obtained through a matching of the marginal distribution. To fulfill this objective, our proposed method harnesses the framework of a Variational Autoencoder (Kingma & Welling, 2013) to learn the recovered latent noise. Specifically, we employ a Gaussian prior distribution for $\mathbf{n}_c$ and $\mathbf{n}_s$ as follows:

$$p_{\mathbf{u}}(\mathbf{n}) = p_{\mathbf{u}}(\mathbf{n}_c)p_{\mathbf{u}}(\mathbf{n}_s) = \mathcal{N}\big(\boldsymbol{\mu}_{\mathbf{n}_c}(\mathbf{u}), \Sigma_{\mathbf{n}_c}(\mathbf{u})\big)\mathcal{N}\big(\boldsymbol{\mu}_{\mathbf{n}_s}(\mathbf{u}), \Sigma_{\mathbf{n}_s}(\mathbf{u})\big), \tag{5}$$

where $\boldsymbol{\mu}$ and $\Sigma$ denote the mean and variance, respectively. Both $\boldsymbol{\mu}$ and $\Sigma$ depend on the domain variable $\mathbf{u}$ and can be implemented with multi-layer perceptrons. The proposed Gaussian prior Eq. 5 gives rise to the following variational posterior:

$$q_{\mathbf{u}}(\mathbf{n}|\mathbf{x}) = q_{\mathbf{u}}(\mathbf{n}_c|\mathbf{x})q_{\mathbf{u}}(\mathbf{n}_s|\mathbf{x}) = \mathcal{N}\big(\boldsymbol{\mu}'_{\mathbf{n}_c}(\mathbf{u}, \mathbf{x}), \Sigma'_{\mathbf{n}_c}(\mathbf{u}, \mathbf{x})\big)\mathcal{N}\big(\boldsymbol{\mu}'_{\mathbf{n}_s}(\mathbf{u}, \mathbf{x}), \Sigma'_{\mathbf{n}_s}(\mathbf{u}, \mathbf{x})\big), \tag{6}$$

where $\boldsymbol{\mu}'$ and $\Sigma'$ denote the mean and variance of the posterior, respectively. Combining the variational posterior with the Gaussian prior, we can derive the following evidence lower bound:

$$\mathcal{L}_{\text{ELBO}} = \mathbb{E}_{q_{\mathbf{u}}(\mathbf{n}|\mathbf{x})}\big(p_{\mathbf{u}}(\mathbf{x})\big) - \beta D_{\mathcal{KL}}\big(q_{\mathbf{u}}(\mathbf{n}|\mathbf{x})||p_{\mathbf{u}}(\mathbf{n})\big), \tag{7}$$

where $D_{\mathcal{KL}}$ denotes the Kullback–Leibler divergence. We here empirically use a hyperparameter $\beta$, motivated by Higgins et al. (2017); Kim & Mnih (2018); Chen et al. (2018), to enhance the independence among $n_i$, considering a common challenge encountered in real-world applications where the availability of source domains is often limited. By maximizing the Evidence Lower Bound (ELBO) as expressed in Eq. 7, we can effectively recover $n_i$ up to scaling and permutation. To address the permutation indeterminacy, as shown in proposition 4.2, we can evaluate the dependence between $\mathbf{y}^{\mathcal{S}}$ and $n_i^{\mathcal{S}}$ to identify which $n_i$ correspond to $\mathbf{n}_c^{\mathcal{S}}$. In the implementation, we use the variational low bounder of mutual information as proposed in Alemi et al. (2016) to quantify the dependence as follows:

$$\mathcal{L}_{\text{MI}} = \mathbb{E}_{q_{\mathbf{u}}(\mathbf{n}_c^{\mathcal{S}}|\mathbf{x})}\big(\log p(\mathbf{y}^{\mathcal{S}}|\mathbf{n}_c^{\mathcal{S}})\big). \tag{8}$$

This loss function serves to maximize the mutual information between $\mathbf{n}_c^{\mathcal{S}}$ and $\mathbf{y}^{\mathcal{S}}$ in source domains. Notably, this maximization also signifies an amplification of the information flow within the causal relationship from $\mathbf{n}_c^{\mathcal{S}}$ for $\mathbf{y}^{\mathcal{S}}$. This alignment between information flow and mutual information arises from the independence among $n_i$, conditioned on $\mathbf{u}$, which is a structural characteristic inherent in the graph representation of our proposed causal model. To promote information flow in the target domain, we can also maximize the mutual information between $\hat{\mathbf{y}}^{\mathcal{T}}$ (representing the estimated label in the target domain) and $\mathbf{n}_c^{\mathcal{T}}$. This is achieved by minimizing the following conditional entropy term:

$$\mathcal{L}_{\text{ENT}} = -\mathbb{E}_{q_{\mathbf{u}}(\mathbf{n}_c^{\mathcal{T}}|\mathbf{x})}\Big(\sum_{\hat{\mathbf{y}}^{\mathcal{T}}} p(\hat{\mathbf{y}}^{\mathcal{T}}|\mathbf{n}_c^{\mathcal{T}})\log p(\hat{\mathbf{y}}^{\mathcal{T}}|\mathbf{n}_c^{\mathcal{T}})\Big), \tag{9}$$

where $\hat{\mathbf{y}}^{\mathcal{T}}$ denotes the estimated label in the target domain. It is interesting to note that this regularization approach has been empirically utilized in previous works to make label predictions more deterministic (Wang et al., 2020; Li et al., 2021). However, our perspective on it differs as we consider it from a causal standpoint. As a result, we arrive at the final loss function:

$$\max \mathcal{L}_{\text{MI}} + \lambda\mathcal{L}_{\text{ELBO}} + \gamma L_{\text{ENT}}, \tag{10}$$

where $\lambda$ and $\gamma$ are hyper-parameters that trade off the three loss functions.

## 6 EXPERIMENTS

**Synthetic Data** We conduct experiments on synthetic data to verify our theoretical results and the ability of the proposed method to adapt to a new domain. Details of synthetic data can be found in Appendix A.2. In the implementation, we use the first 4 segments as source domains, and the last segment as the target domain. Figure 3(a) shows the true and recovered distributions of $\mathbf{n}_c$. The proposed iLCC-LCS obtains the mean correlation coefficient (MCC) of 0.96 between the original $\mathbf{n}_c$ and the recovered one. Due to the invariant conditional distribution $p(\mathbf{y}|\mathbf{n}_c)$, even with the change of $p(\mathbf{n}_c)$ as shown in Figure 3(a), the learned $p(\mathbf{y}|\mathbf{n}_c)$ can generalize to target segment in a principle way as depicted by the Figure 3(b).

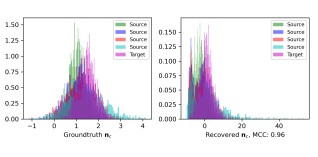 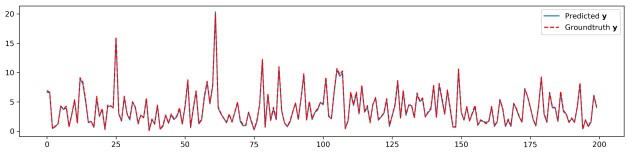

   (a) Recovered $\mathbf{n}_c$          (b) Predicted $\mathbf{y}$ on the target segment

Figure 3: The Result on Synthetic Data.

Table 1: Ablation study on resampled PACS data ($D_{\mathcal{KL}} = 0.7$), and TerraIncognita.

| Methods | Accuracy on resampled PACS data($D_{\mathcal{KL}} = 0.7$) | | | | | Accuracy on TerraIncognita | | | | |
|---|---|---|---|---|---|---|---|---|---|---|
| | →Art | →Cartoon | →Photo | →Sketch | Average | →L28 | →L43 | →L46 | →L7 | Average |
| iLCC-LCS with $\beta = 1$ | 90.2 ± 0.5 | 73.4 ± 0.8 | 95.7 ± 0.4 | 82.7 ± 0.7 | 85.5 | 56.3 ± 4.3 | 61.5 ± 0.7 | 45.2 ± 0.3 | 80.1 ± 0.6 | 60.8 |
| iLCC-LCS with $\gamma = 0$ | 81.1 ± 1.5 | 70.0 ± 1.6 | 92.0 ± 0.5 | 59.6 ± 0.7 | 75.7 | 54.8 ± 1.4 | 58.9 ± 1.8 | **46.8 ± 1.4** | 73.1 ± 0.6 | 58.4 |
| iLCC-LCS | **90.7 ± 0.3** | 74.2 ± 0.7 | **95.8 ± 0.3** | **83.0 ± 2.2** | **86.0** | **64.3 ± 3.4** | **63.1 ± 1.6** | 44.7 ± 0.4 | **80.8 ± 0.4** | **63.2** |

**Resampled PACS data:** There exist datasets, such as PACS (Li et al., 2017) and Office-Home (Venkateswara et al., 2017), commonly used for evaluating MSDA under previous paradigms. These datasets exhibit very limited changes in label distribution across domains. For instance, in the PACS dataset, the KL divergence of label distributions between any two domains is exceedingly small, approximately $D_{\mathcal{KL}} \approx 0.1$. Consequently, these datasets are well-suited for evaluating models in the setting of conditional shift, where label distributions remain consistent across domains, as illustrated in Figure 1(b). In order to render the PACS dataset more suitable for assessing the adaptability of MSDA algorithms to more challenging and realistic scenarios characterized by significant label distribution shifts, we apply a filtering process. This involves random sampling from the original PACS dataset to obtain a predefined label distribution. As a result, we generate three new datasets: PACS ($D_{\mathcal{KL}} = 0.3$), PACS ($D_{\mathcal{KL}} = 0.5$), and PACS ($D_{\mathcal{KL}} = 0.7$). Here, $D_{\mathcal{KL}} = 0.3, 0.5, 0.7$ indicates that the KL divergence of label distributions between any two distinct domains is approximately 0.3, 0.5, 0.7. Further elaboration on the label distribution across domains can be found in Appendix A.1.

**Terra Incognita data:** We further evaluate the proposed iLCC-LCS on Terra Incognita dataset proposed in (Beery et al., 2018) used for evaluation for domain generalization. In this dataset, the label distribution is long-tailed at each domain, and each domain has a different label distribution, hence it naturally has significant label distribution shifts, ideal for the challenging scenarios that LCS describes. We select four domains from the original data, L28, L43, L46, and L7, which share the same seven categories: bird, bobcat, empty, opossum, rabbit, raccoon, and skunk. Here 'L28' denotes that the image data is collected from location 28. More detailed label distribution across domains can be found in Appendix A.1.

**Baselines** We compare the proposed method with state-of-the-art methods to verify its effectiveness. Particularly, we compare the proposed methods with empirical risk minimization (ERM), MCDA (Saito et al., 2018), M3DA (Peng et al., 2019), LtC-MSDA (Wang et al., 2020), T-SVDNet (Li et al., 2021), IRM (Arjovsky et al., 2019), IWCDAN (Tachet des Combes et al., 2020) and LaCIM (Sun et al., 2021). In these methods, MCDA, M3DA, LtC-MSDA and T-SVDNet learn invariant representations for MSDA, while IRM, IWCDAN and LaCIM are tailored for label distribution shifts. Details of implementation, including network architectures and hyper-parameter settings, are in APPENDIX A.3. All the methods above are averaged over 3 runs.

**Ablation studies** Table 1 displays the results of our ablation studies conducted on PACS ($D_{\mathcal{KL}} = 0.3$) and TerraIncognita, respectively. Notably, we observe a significant improvement in performance (approximately 10% and 5%, respectively) for the proposed method on both datasets when employing entropy regularization (as per Eq. 9). This finding aligns with previous research (Wang et al., 2020; Li et al., 2021), which has also underscored the importance of entropy regularization (Eq. 9). From the viewpoint of the proposed causal model, entropy regularization essentially encourages causal influence between $\mathbf{y}$ and $\mathbf{n}_c$, as elaborated upon in Section 5. Furthermore, the hyperparameter $\beta$ plays a pivotal role in enhancing performance by enforcing independence among the latent variables $n_i$.

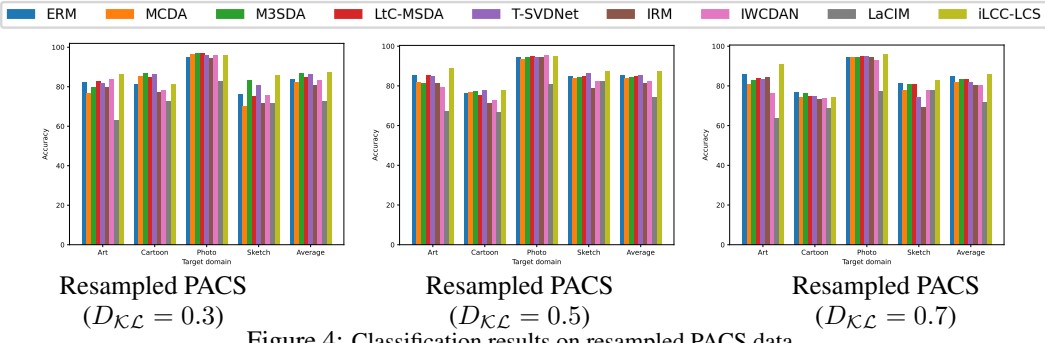

Figure 4: Classification results on resampled PACS data.

Table 2: Classification results on TerraIncognita.

| Methods | Accuracy | | | | |
|---|---|---|---|---|---|
| | →L28 | →L43 | →L46 | →L7 | Average |
| ERM | $54.1 \pm 2.8$ | $62.3 \pm 0.7$ | $44.7 \pm 0.9$ | $74.5 \pm 2.6$ | 58.9 |
| MCDA (Saito et al., 2018) | $54.9 \pm 4.1$ | $61.2 \pm 1.2$ | $42.7 \pm 0.3$ | $64.8 \pm 8.1$ | 55.9 |
| M3SDA (Peng et al., 2019) | $62.3 \pm 1.4$ | $62.7 \pm 0.4$ | $41.3 \pm 0.3$ | $57.4 \pm 0.9$ | 55.9 |
| LtC-MSDA (Wang et al., 2020) | $51.9 \pm 5.7$ | $54.6 \pm 1.3$ | $45.7 \pm 1.0$ | $69.1 \pm 0.3$ | 55.3 |
| T-SVDNet (Li et al., 2021) | $58.2 \pm 1.7$ | $61.9 \pm 0.3$ | $45.6 \pm 2.0$ | $68.2 \pm 1.1$ | 58.5 |
| IRM (Arjovsky et al., 2019) | $57.5 \pm 1.7$ | $60.7 \pm 0.3$ | $42.4 \pm 0.6$ | $74.1 \pm 1.6$ | 58.7 |
| IWCDAN (Tachet des Combes et al., 2020) | $58.1 \pm 1.8$ | $59.3 \pm 1.9$ | $43.8 \pm 1.5$ | $58.9 \pm 3.8$ | 55.0 |
| LaCIM (Sun et al., 2021) | $58.2 \pm 3.3$ | $59.8 \pm 1.6$ | $46.3 \pm 1.1$ | $70.8 \pm 1.0$ | 58.8 |
| iLCC-LCS(Ours) | $\mathbf{64.3 \pm 3.4}$ | $\mathbf{63.1 \pm 1.6}$ | $44.7 \pm 0.4$ | $\mathbf{80.8 \pm 0.4}$ | $\mathbf{63.2}$ |

**Results** Due to limited space, the results by different methods on the resampled PACS are presented in Figure 4. Detailed results can be found in Tables 3-5 in Appendix A.4. We can observe that as the increase of KL divergence of label distribution, the performances of MCDA, M3DA, LtC-MSDA and T-SVDNet, which are based on learning invariant representations, gradually degenerate. In the case where the KL divergence is about 0.7, the performances of these methods are worse than traditional ERM. Compared with IRM, IWCDAN and LaCIM, specifically designed for label distribution shifts, the proposed iLCC-LCS obtains the best performance, due to our theoretical guarantee for identifying the latent causal content variable, resulting in a principled way to guarantee adaptation to the target domain. Table 2 depicts the results by different methods on challenging Terra Incognit. The proposed iLCC-LCS achieves a significant performance gain on the challenging task →L7. Among the methods considered, the proposed proposed iLCC-LCS stands out as the only one that outperforms ERM. This superiority is especially pronounced due to the substantial variation in label distribution across domains (see in APPENDIX A.3 for details of label distribution). In cases where the label distribution changes significantly, traditional approaches may lead to the development of uninformative features, making them less effective for domain adaptation. In contrast, the proposed method excels at capturing and adapting to these label distribution changes, enabling accurate predictions even under such dynamic conditions.

# 7  CONCLUSION

The key for domain adaptation is to understand how the joint distribution of features and label changes across domains. Previous works usually assume covariate shift or conditional shift to interpret the change of the joint distribution, which may be restricted in some real applications that label distribution shifts. Hence, this work considers a new and milder assumption, latent covariate shift. Specifically, we propose a latent causal model to precisely formulate the generative process of input features and labels. We show that the latent content variable in the proposed latent causal model can be identified up to scaling. This inspires a new method to learn the invariant label distribution conditional on the latent causal variable, resulting in a principled way to guarantee adaptation to target domains. Experiments demonstrate the theoretical results and the efficacy of the proposed method, compared with state-of-the-art methods across various data sets.

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

# A   APPENDIX

## A.1   DATA DETAILS

We resample the original PACS dataset, which contains 4 domains, Photo, Artpainting, Cartoon, and Sketch, and shares the same seven categories, since the KL divergence of label distributions of any two domains in the original PACS are very similar, *e.g.*, $D_{\mathcal{KL}} \approx 0.1$. To make PACS dataset suitable for evaluation in the setting of the proposed LCS, we filter the original PACS dataset by re-sampling it, *i.e.*, we randomly filter some samples in the original PACS to obtain pre-defined label distribution. As a result, we can obtain three new datasets, resampled PACS ($D_{\mathcal{KL}} = 0.3$), resampled PACS ($D_{\mathcal{KL}} = 0.5$), and resampled PACS ($D_{\mathcal{KL}} = 0.7$). Here $D_{\mathcal{KL}} = 0.3(0.5, 0.7)$ denotes that KL divergence of label distributions in any two different domains is approximately 0.3 (0.5, 0.7). Figure 5 depicts label distributions in these resampled datasets.

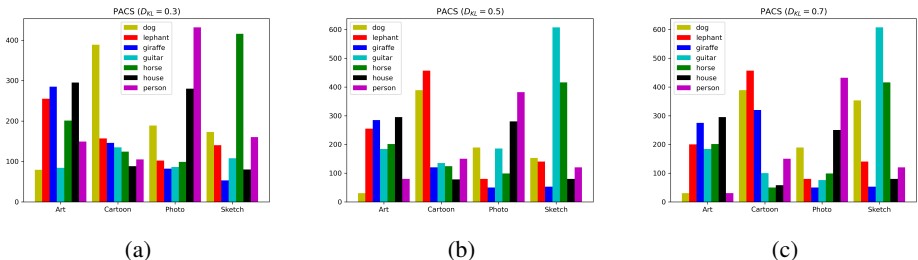

(a)                                   (b)                                   (c)

Figure 5: Label distributions of the resampled PACS datasets with $D_{\mathcal{KL}} = 0.3$ (a), $D_{\mathcal{KL}} = 0.5$ (b), $D_{\mathcal{KL}} = 0.7$ (c).

For Terra Incognita (Beery et al., 2018), it consists of 57, 868 images across 20 locations, each labeled with one of 15 classes (or marked as empty). Classes are either single species (e.g. "Coyote" or groups of species, e.g. "Bird"). Figure 6 shows the label distribution in different locations. The label distribution is long-tailed at each domain, and each domain has a different label distribution, which is naturally applicable to our setting. We use the four domains from the original data, L28, L43, L46, and L7, which share the same seven categories: bird, bobcat, empty, opossum, rabbit, raccoon, and skunk. Here 'L28' denotes that the image data is collected from location 28. Figure 6(b) depicts label distributions in the four domains above.

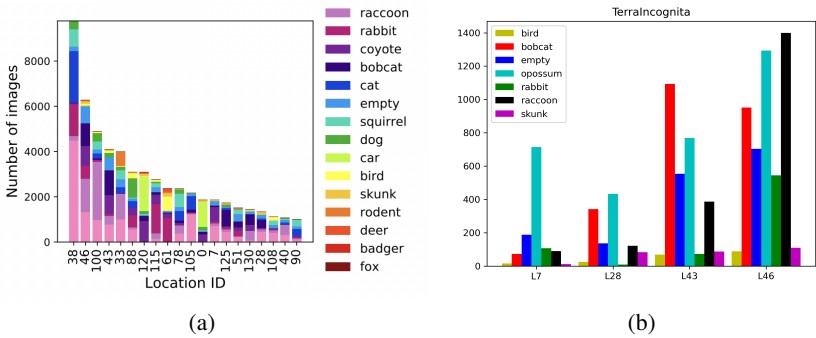

(a)                                                     (b)

Figure 6: (a) Label distributions of the whole Terra Incognita data. (b) Label distributions in the four domains of the Terra Incognita data, which are used in our experiments.

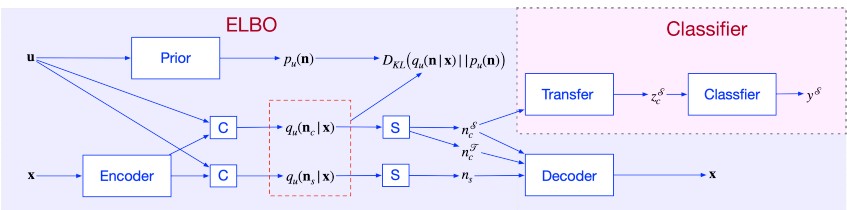

Figure 7: The proposed iLCC-LCS to learn the invariant $p(\mathbf{y}|\mathbf{n}_c)$ for multiple source domain adaptation. C denotes concatenation, and S denotes sampling from the posterior distributions.

## A.2 SYNTHETIC DATA

The synthetic data generative process is as follows: we divide the latent variables into 5 segments, which correspond to 5 domains. Each segment includes 1000 examples. Within each segment, we first sample the mean and the variance from uniform distributions $[1, 2]$ and $[0.3, 1]$ for the latent exogenous variables $\mathbf{n}_c$ and $\mathbf{n}_s$, respectively. Then for each segment, we generate $\mathbf{z}_c$, $\mathbf{z}_s$, $\mathbf{x}$ and $\mathbf{y}$ according to the following structural causal model:

$$\mathbf{z}_c := \mathbf{n}_c, \mathbf{z}_s := \mathbf{z}_c^3 + \mathbf{n}_s, \mathbf{y} := \mathbf{z}_c^3, \mathbf{x} := \mathrm{MLP}(\mathbf{z}_c, \mathbf{z}_s), \tag{11}$$

where following (Khemakhem et al., 2020) we mix the latent $\mathbf{z}_c$ and $\mathbf{z}_s$ using a multi-layer perceptron to generate $\mathbf{x}$.

## A.3 IMPLEMENTATION DETAILS

For the synthetic data, we used an encoder, *e.g.* 3-layer fully connected network with 30 hidden nodes for each layer, and a decoder, *e.g.* 3-layer fully connected network with 30 hidden nodes for each layer. We use a 3-layer fully connected network with 30 hidden nodes for the prior model. Since this is an ideal environment to verify the proposed method, for hyper-parameters, we set $\beta = 1$ and $\gamma = 0$ to remove the heuristic constraints, and we set $\lambda = 1e - 2$. For the real data, all methods used the same network backbone, ResNet-18 pre-trained on ImageNet. Since it can be challenging to train VAE on high-resolution images, we use extracted features by ResNet-18 as our VAE input. We then use 2-layer fully connected networks as the VAE encoder and decoder, use 2-layer fully connected network for the prior model, and use 2-layer fully connected network to transfer $\mathbf{n}_c$ to $\mathbf{z}_c$. For hyper-parameters, we set $\beta = 4$, $\gamma = 0.1$, $\lambda = 1e - 4$ for the proposed method on all datasets. A graphical depiction of the proposed iLCC-MSDA is shown in Figure 7.

Table 3: Classification results on resampled PACS data ($D_{\mathcal{KL}} = 0.3$).

| Methods | Accuracy | | | | |
|---|---|---|---|---|---|
| | →Art | →Cartoon | →Photo | →Sketch | Average |
| ERM | $82.3 \pm 0.3$ | $81.3 \pm 0.9$ | $94.9 \pm 0.2$ | $76.2 \pm 0.7$ | 83.6 |
| MCDA ((Saito et al., 2018)) | $76.6 \pm 0.6$ | $85.1 \pm 0.3$ | $96.6 \pm 0.1$ | $70.1 \pm 1.3$ | 82.1 |
| M3SDA (Peng et al., 2019) | $79.6 \pm 1.0$ | $\mathbf{86.6 \pm 0.5}$ | $\mathbf{97.1 \pm 0.3}$ | $83.3 \pm 1.0$ | 86.6 |
| LtC-MSDA (Wang et al., 2020) | $82.7 \pm 1.3$ | $84.9 \pm 1.4$ | $96.9 \pm 0.2$ | $75.3 \pm 3.1$ | 84.9 |
| T-SVDNet (Li et al., 2021) | $81.8 \pm 0.3$ | $86.5 \pm 0.2$ | $95.9 \pm 0.2$ | $80.7 \pm 0.8$ | 86.3 |
| IRM (Arjovsky et al., 2019) | $79.6 \pm 0.7$ | $77.0 \pm 2.2$ | $94.6 \pm 0.2$ | $71.7 \pm 2.3$ | 80.7 |
| IWCDAN (Tachet des Combes et al., 2020) | $84.0 \pm 0.5$ | $78.1 \pm 0.7$ | $96.0 \pm 0.1$ | $75.5 \pm 1.9$ | 83.4 |
| LaCIM (Sun et al., 2021) | $63.1 \pm 1.5$ | $72.6 \pm 1.0$ | $82.7 \pm 1.3$ | $71.5 \pm 0.9$ | 72.5 |
| iLCC-LCS(Ours) | $\mathbf{86.4 \pm 0.8}$ | $81.1 \pm 0.8$ | $95.9 \pm 0.1$ | $\mathbf{86.0 \pm 1.0}$ | $\mathbf{87.4}$ |

Table 4: Classification results on resampled PACS data ($D_{\mathcal{KL}} = 0.5$).

| Methods | Accuracy | | | | |
|---|---|---|---|---|---|
| | →Art | →Cartoon | →Photo | →Sketch | Average |
| ERM | $85.4 \pm 0.6$ | $76.4 \pm 0.5$ | $94.4 \pm 0.4$ | $85.0 \pm 0.6$ | 85.3 |
| MCDA ((Saito et al., 2018)) | $81.6 \pm 0.1$ | $76.8 \pm 0.1$ | $93.6 \pm 0.1$ | $84.1 \pm .6$ | 84.0 |
| M3SDA (Peng et al., 2019) | $81.2 \pm 1.2$ | $77.5 \pm 1.3$ | $94.5 \pm 0.5$ | $84.3 \pm 0.5$ | 84.4 |
| LtC-MSDA (Wang et al., 2020) | $85.2 \pm 1.5$ | $75.2 \pm 2.6$ | $94.9 \pm 0.6$ | $85.1 \pm 2.7$ | 85.1 |
| T-SVDNet (Li et al., 2021) | $84.8 \pm 0.3$ | $77.6 \pm 1.7$ | $94.2 \pm 0.2$ | $86.4 \pm 0.2$ | 85.6 |
| IRM (Arjovsky et al., 2019) | $81.5 \pm 0.3$ | $71.1 \pm 1.3$ | $94.2 \pm 0.1$ | $78.7 \pm 0.7$ | 81.4 |
| IWCDAN (Tachet des Combes et al., 2020) | $79.2 \pm 1.6$ | $72.6 \pm 0.7$ | $\mathbf{95.6 \pm 0.1}$ | $82.1 \pm 2.2$ | 82.4 |
| LaCIM (Sun et al., 2021) | $67.4 \pm 1.6$ | $66.6 \pm 0.6$ | $81.0 \pm 1.2$ | $82.3 \pm 0.6$ | 74.3 |
| iLCC-LCS(Ours) | $\mathbf{89.0 \pm 0.7}$ | $\mathbf{77.6 \pm 0.5}$ | $95.0 \pm 0.3$ | $\mathbf{87.4 \pm 1.6}$ | $\mathbf{87.3}$ |

## A.4 CLASSIFICATION RESULTS ON RESAMPLED PACS DATA

Since the KL divergence of label distributions between any two domains in original PACS data (Li et al., 2017) is exceedingly small, approximately $D_{\mathcal{KL}} \approx 0.1$. We apply a filtering process, which involves random sampling from the original PACS dataset to obtain a predefined label distribution. As a result, we generate three new datasets: PACS ($D_{\mathcal{KL}} = 0.3$), resampled PACS ($D_{\mathcal{KL}} = 0.5$), and resampled PACS ($D_{\mathcal{KL}} = 0.7$). Here $D_{\mathcal{KL}} = 0.3(0.5, 0.7)$ denotes that KL divergence of label distributions in any two different domains is approximately 0.3 (0.5, 0.7).

We evaluate the different methods on the resampled PACS data, and results can be found in Tables 3-5. We can see that the proposed method performs better, compared with state-of-the-art methods. In scenarios marked by substantial shifts in label distribution, conventional methods can inadvertently give rise to uninformative features, undermining their efficacy in domain adaptation. Conversely, our proposed method excels in capturing and adapting to these dynamic label distribution changes, empowering it to make accurate predictions even under such challenging conditions.

Table 5: Classification results on resampled PACS data ($D_{\mathcal{KL}} = 0.7$).

| Methods | Accuracy | | | | |
|---|---|---|---|---|---|
| | →Art | →Cartoon | →Photo | →Sketch | Average |
| ERM | $86.1 \pm 0.6$ | $\mathbf{76.8 \pm 0.3}$ | $94.6 \pm 0.4$ | $81.3 \pm 2.0$ | 84.7 |
| MCDA ((Saito et al., 2018)) | $80.8 \pm 0.7$ | $74.1 \pm 1.2$ | $94.4 \pm 0.4$ | $77.9 \pm 0.4$ | 81.8 |
| M3SDA (Peng et al., 2019) | $82.7 \pm 1.3$ | $76.2 \pm 1.0$ | $94.5 \pm 0.7$ | $80.8 \pm 1.2$ | 83.6 |
| LtC-MSDA (Wang et al., 2020) | $83.7 \pm 1.6$ | $74.6 \pm 1.4$ | $95.0 \pm 0.7$ | $80.8 \pm 0.6$ | 83.5 |
| T-SVDNet (Li et al., 2021) | $83.3 \pm 0.8$ | $74.7 \pm 0.6$ | $95.2 \pm 0.3$ | $74.5 \pm 3.3$ | 81.9 |
| IRM (Arjovsky et al., 2019) | $84.3 \pm 0.8$ | $73.3 \pm 1.8$ | $94.3 \pm 0.1$ | $69.4 \pm 4.6$ | 80.3 |
| IWCDAN (Tachet des Combes et al., 2020) | $76.3 \pm 0.8$ | $73.9 \pm 1.6$ | $93.1 \pm 0.5$ | $77.6 \pm 3.8$ | 80.2 |
| LaCIM (Sun et al., 2021) | $63.6 \pm 0.9$ | $68.7 \pm 1.4$ | $77.5 \pm 3.8$ | $77.8 \pm 2.2$ | 71.9 |
| iLCC-LCS(Ours) | $\mathbf{90.7 \pm 0.3}$ | $74.2 \pm 0.7$ | $\mathbf{95.8 \pm 0.3}$ | $83.0 \pm 2.2$ | $\mathbf{86.0}$ |

## A.5 T-SNE VISUALIZATION

We visualize features of each class obtained by the proposed method via t-SNE to show how the distributions of learned features change across domains. Figure 8 (a)-(d) depicts the detailed distributions of the learned features by the proposed method in different 4 domains of TerraIncognita data. Differing from the methods that learn invariant representations across domains, the distributions of the learned features by the proposed method change across domains.

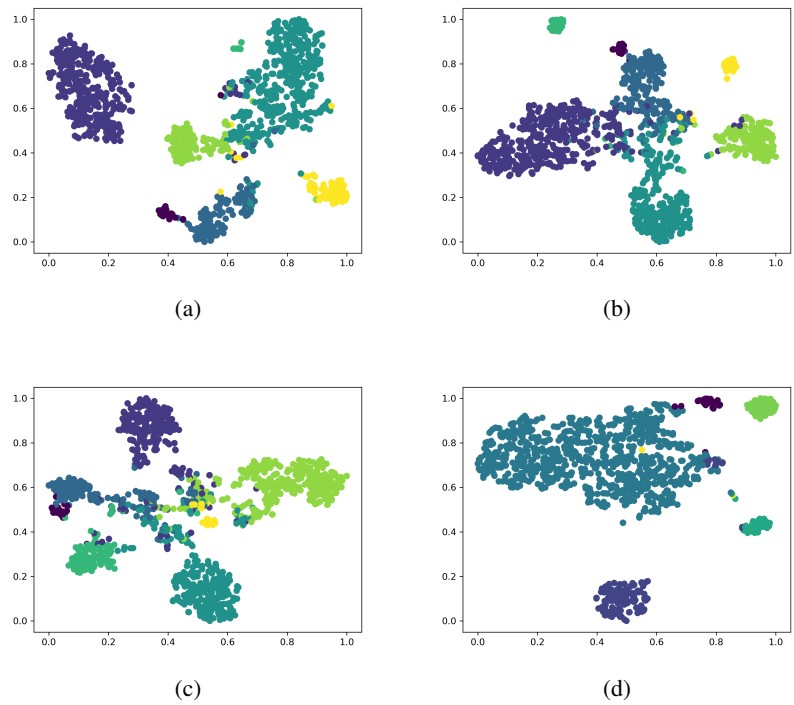

(a)                    (b)

(c)                    (d)

Figure 8: The t-SNE visualizations of learned features $\mathbf{n}_c$ of different domains on the $\rightarrow$L7 task in TerraIncognita. (a)-(d) The learned features $\mathbf{n}_c$ in domain L28, L43, L46, L7. Here 'L28' denotes that the image data is collected from location 28. We can observe that the distribution of learned feature $\mathbf{n}_c$ by the proposed method changes across domains.

## A.6 THE PROOF OF PROPOSITION 4.1

To establish non-identifiability, it suffices to demonstrate the existence of an alternative solution that differs from the ground truth but can produce the same observed data. Let's consider the latent causal generative model defined in Eqs. 1 to 3 as our ground truth, represented in Figure 2(a), where there exists a causal relationship from $\mathbf{z}_c$ to $\mathbf{z}_s$. Now, we can always construct new latent variables $\mathbf{z}'$ as follows: $\mathbf{z}'_c = \mathbf{g}_c(\mathbf{n}_c)$ and $\mathbf{z}'_s = \mathbf{n}_s$, depicted in Figure 2(b). Importantly, there is no causal influence from $\mathbf{z}'_c$ to $\mathbf{z}'_s$ in this construction, which is different from the ground truth in Figure 2(a). It becomes evident that this new set of latent variables $\mathbf{z}'$ can generate the same $\mathbf{x}$ as obtained by $\mathbf{z}$ through a composition function $\mathbf{f} \circ \mathbf{f}'$, where $\mathbf{f}'(\mathbf{z}') = [\mathbf{z}'_c, \mathbf{g}_{s2}(\mathbf{g}_{s1}(\mathbf{z}'_c) + \mathbf{z}'_s)]$. This scenario leads to a non-identifiable result, as the same observed data can be produced by different latent variable configurations, thus confirming non-identifiability.

## A.7 THE PROOF OF PROPOSITION 4.2

For convenience, we first introduce the following lemma.

**Lemma A.1** *Denote the mapping from $\mathbf{n}$ to $\mathbf{z}$ by $\mathbf{g}$, given the assumption (ii) in proposition 4.2 that the mapping $\mathbf{f}$ from $\mathbf{z}$ to $\mathbf{x}$ is invertible, we have that the mapping from $\mathbf{n}$ to $\mathbf{x}$, e.g., $\mathbf{f} \circ \mathbf{g}$, is invertible.*

Proof can be easily shown by the following: since the mapping $\mathbf{g}$ from $\mathbf{n}$ to $\mathbf{z}$ is defined as 2 where both $\mathbf{g}_c$ and $\mathbf{g}_{s2}$ are assumed to be invertible, we can obtain the inverse function of the mapping $\mathbf{g}$ from $\mathbf{n}$ to $\mathbf{z}$ as follows: $\mathbf{n}_c = \mathbf{g}_c^{-1}(\mathbf{z}_c)$, $\mathbf{n}_s = \mathbf{g}_{s2}^{-1}(\mathbf{z}_s) - \mathbf{g}_{s1}^{-1}(\mathbf{z}_c)$, which clearly shows that the the mapping $\mathbf{g}$ is invertible. Then by the assumption (ii) in proposition 4.2 that $\mathbf{f}$ is invertible, we have that the composition of $\mathbf{f}$ and $\mathbf{g}$ is invertible, i.e., $\mathbf{f} \circ \mathbf{g}$ is invertible.

The proof of proposition 4.2 is done in three steps. Step I is to show that the latent noise variables $\mathbf{n}$ can be identified up to linear transformation, $\mathbf{n} = \mathbf{A}\hat{\mathbf{n}} + \mathbf{c}$, where $\hat{\mathbf{n}}$ denotes the recovered latent noise variable obtained by matching the true marginal data distribution. Step II shows that the linear transformation can be reduced to permutation transformation, i.e., $\mathbf{n} = \mathbf{P}(\hat{\mathbf{n}}) + \mathbf{c}$. Step III shows that $\mathbf{z}_c = \mathbf{h}(\hat{\mathbf{z}}_c)$, by using d-separation criterion in the graph structure in Figure 2(a), i.e., $\mathbf{y}^{\mathcal{S}}$ is dependent on $\mathbf{n}_c^{\mathcal{S}}$ but independent of $\mathbf{n}_s^{\mathcal{S}}$, given $\mathbf{u}$.

**Step I:** Suppose we have two sets of parameters $\theta = (\mathbf{f}, \mathbf{g}, \mathbf{T}, \boldsymbol{\eta})$ and $\theta = (\hat{\mathbf{f}}, \hat{\mathbf{g}}, \hat{\mathbf{T}}, \hat{\boldsymbol{\eta}})$ corresponding to the same conditional probabilities, i.e., $p_{(\mathbf{f},\mathbf{g},\mathbf{T},\boldsymbol{\eta})}(\mathbf{x}|\mathbf{u}) = p_{((\hat{\mathbf{f}},\hat{\mathbf{g}},\hat{\mathbf{T}},\hat{\boldsymbol{\eta}}))}(\mathbf{x}|\mathbf{u})$ for all pairs $(\mathbf{x}, \mathbf{u})$. Since the mapping from $\mathbf{n}$ to $\mathbf{x}$ is invertible, as Lemma A.1, with the assumption (i), by expanding the conditional probabilities (see Step I for proof of Theorem 1 in Khemakhem et al. (2020) for more details), we have:

$$\log|\det \mathbf{J}_{(\mathbf{f}\circ\mathbf{g})^{-1}}(\mathbf{x})| + \log p_{(\mathbf{T},\boldsymbol{\eta})}(\mathbf{n}|\mathbf{u}) = \log|\det \mathbf{J}_{(\hat{\mathbf{f}}\circ\hat{\mathbf{g}})^{-1}}(\mathbf{x})| + \log p_{(\hat{\mathbf{T}},\hat{\boldsymbol{\eta}})}(\hat{\mathbf{n}}|\mathbf{u}), \quad (12)$$

Using the exponential family Eq. 1 to replace $p_{(\mathbf{T_n},\boldsymbol{\beta})}(\mathbf{n}|\mathbf{u})$, we have:

$$\log|\det \mathbf{J}_{(\mathbf{f}\circ\mathbf{g})^{-1}}(\mathbf{x})| + \mathbf{T}^T\big((\mathbf{f}\circ\mathbf{g})^{-1}(\mathbf{x})\big)\boldsymbol{\eta}(\mathbf{u}) - \log\prod_i Z_i(\mathbf{u}) =$$

$$\log|\det \mathbf{J}_{(\hat{\mathbf{f}}\circ\hat{\mathbf{g}})^{-1}}(\mathbf{x})| + \hat{\mathbf{T}}^T\big((\hat{\mathbf{f}}\circ\hat{\mathbf{g}})^{-1}(\mathbf{x})\big)\hat{\boldsymbol{\eta}}(\mathbf{u}) - \log\prod_i \hat{Z}_i(\mathbf{u}), \quad (13)$$

Then by expanding the above at points $\mathbf{u}_l$ and $\mathbf{u}_0$ mentioned in assumption (iii), and using Eq. 13 at point $\mathbf{u}_l$ subtract Eq. 13 at point $\mathbf{u}_0$, we find:

$$\langle \mathbf{T}(\mathbf{n}), \bar{\boldsymbol{\eta}}(\mathbf{u}) \rangle + \sum_i \log \frac{Z_i(\mathbf{u}_0)}{Z_i(\mathbf{u}_l)} = \langle \hat{\mathbf{T}}(\hat{\mathbf{n}}), \bar{\hat{\boldsymbol{\eta}}}(\mathbf{u}) \rangle + \sum_i \log \frac{\hat{Z}_i(\mathbf{u}_0)}{\hat{Z}_i(\mathbf{u}_l)}. \quad (14)$$

Here $\bar{\boldsymbol{\eta}}(\mathbf{u}_l) = \boldsymbol{\eta}(\mathbf{u}_l) - \boldsymbol{\eta}(\mathbf{u}_0)$. Then by combining the $2\ell$ expressions (from assumption (iii) we have $2\ell$ such expressions) into a single matrix equation, we can write this in terms of $\mathbf{L}$ from assumption (iii),

$$\mathbf{L}^T\mathbf{T}(\mathbf{n}) = \hat{\mathbf{L}}^T\hat{\mathbf{T}}(\hat{\mathbf{n}}) + \mathbf{b}. \quad (15)$$

Since $\mathbf{L}^T$ is invertible, we can multiply this expression by its inverse from the left to get:

$$\mathbf{T}(\mathbf{n}) = \mathbf{A}\hat{\mathbf{T}}(\hat{\mathbf{n}}) + \mathbf{c}, \quad (16)$$

where $\mathbf{A} = (\mathbf{L}^T)^{-1}\hat{\mathbf{L}}^{\mathbf{T}}$. According to a lemma 3 in Khemakhem et al. (2020) that there exist $k$ distinct values $n_i^1$ to $n_i^k$ such that the Derivative $T'(n_i^1), ..., T'(n_i^k)$ are linearly independent, and the fact that each component of $T_{i,j}$ is univariate. We can show that $\mathbf{A}$ is invertible.

**Step II** Since we assume the noise to be two-parameter exponential family members, Eq. 16 can be re-expressed as:

$$\left( \begin{array}{c} \mathbf{T}_1(\mathbf{n}) \\ \mathbf{T}_2(\mathbf{n}) \end{array} \right) = \mathbf{A} \left( \begin{array}{c} \hat{\mathbf{T}}_1(\hat{\mathbf{n}}) \\ \hat{\mathbf{T}}_2(\hat{\mathbf{n}}) \end{array} \right) + \mathbf{c}, \tag{17}$$

Then, we re-express $\mathbf{T}_2$ in term of $\mathbf{T}_1$, e.g., $T_2(n_i) = t(T_1(n_i))$ where $t$ is a nonlinear mapping. As a result, we have from Eq. 17 that: (a) $T_1(n_i)$ can be linear combination of $\hat{\mathbf{T}}_1(\hat{\mathbf{n}})$ and $\hat{\mathbf{T}}_2(\hat{\mathbf{n}})$, and (b) $t(T_1(n_i))$ can also be linear combination of $\hat{\mathbf{T}}_1(\hat{\mathbf{n}})$ and $\hat{\mathbf{T}}_2(\hat{\mathbf{n}})$. This implies the contradiction that both $T_1(n_i)$ and its nonlinear transformation $t(T_1(n_i))$ can be expressed by linear combination of $\hat{\mathbf{T}}_1(\hat{\mathbf{n}})$ and $\hat{\mathbf{T}}_2(\hat{\mathbf{n}})$. This contradiction leads to that $\mathbf{A}$ can be reduced to permutation matrix $\mathbf{P}$ (See APPENDIX C in Sorrenson et al. (2020) for more details):

$$\mathbf{n} = \mathbf{P}\hat{\mathbf{n}} + \mathbf{c}, \tag{18}$$

where $\mathbf{P}$ denote the permutation matrix with scaling, $\mathbf{c}$ denote a constant vector. Note that this result holds for not only Gaussian, but also inverse Gaussian, Beta, Gamma, and Inverse Gamma (See Table 1 in Sorrenson et al. (2020)).

**Step III:** The above result shows that we can obtain the recovered latent noise variables $\hat{\mathbf{n}}$ up to permutation and scaling transformation of the true $\mathbf{n}$, which are learned by matching the true marginal data distribution $p(\mathbf{x}|\mathbf{u})$. However, it is still not clear which part of $\hat{\mathbf{n}}$ corresponds to $\mathbf{n}_c$, e.g., permutation indeterminacy. Fortunately, the permutation can be removed by d-separation criteria in the graph structure in Figure 2(a), which implies that $\mathbf{n}_c$ is dependent on label $\mathbf{y}$, while $\mathbf{n}_s$ is independent with $\mathbf{y}$, given $\mathbf{u}$. Thus, together with the d-separation criteria, Eq. 18 implies that:

$$\mathbf{n}_c = \mathbf{P}_c\hat{\mathbf{n}}_c + \mathbf{c}_c. \tag{19}$$

Then by model assumption as defined in Eq. 2, using $\mathbf{z}_c$ to replace $\mathbf{n}_c$ obtains:

$$\mathbf{g}_c^{-1}(\mathbf{z}_c) = \mathbf{P}_c\hat{\mathbf{g}}_c^{-1}(\hat{\mathbf{z}}_c) + \mathbf{c}_c, \tag{20}$$

which can re-expressed as:

$$\mathbf{z}_c = \mathbf{g}_c(\mathbf{P}_c^{-1}(\hat{\mathbf{g}}_c^{-1}(\hat{\mathbf{z}}_c) - \mathbf{c}_c)) = \mathbf{h}((\hat{\mathbf{z}}_c)), \tag{21}$$

where $\mathbf{h}$ is clearly invertible, which implies that $\mathbf{z}_c$ can be recovered up to block identifiability.

## A.8 UNDERSTANDING ASSUMPTIONS FOR IDENTIFYING $\mathbf{z}_c$

Assumptions (i)-(iii) are motivated by the nonlinear ICA literature (Khemakhem et al., 2020), which is to provide a guarantee that we can recover latent noise variables $\mathbf{n}$ up to a permutation and scaling transformation. The main Assumption (iii) essentially requires sufficient changes in latent noise variables to facilitate their identification.

Furthermore, we conduct model assumptions, as defined in Eqs. (1)-(3). Essentially, Eq. (1), derived from Sorrenson et al. (2020), posits that each $n_i$ is sampled from the two-parameter exponential family. We adopt this assumption in consideration of real-world applications where the dimension of $n_i$ is unknown. By assuming two-parameter exponential family members, it has been demonstrated that informative latent noise variables $n_i$ can be automatically separated from noise by an estimating model (for more details, refer to Sorrenson et al. (2020)). However, it is important to acknowledge that in real applications, the distribution of $\mathbf{n}$ could be arbitrary. In this context, assumption Eq. (1) may only serve as an approximation for the true distribution of $\mathbf{n}$. Nevertheless, in terms of the performance of the proposed method on real datasets such as PACS and TerraIncognita, such an approximation may be deemed acceptable to some extent. To establish a coherent connection between these latent noise variables $\mathbf{n}$ and the latent variables $\mathbf{z}$, we assume post-nonlinear models, as defined in Eq. 2. We posit that post-nonlinear models could be further relaxed, as long as the assumed models are invertible from $\mathbf{n}$ to $\mathbf{x}$. Here, we choose to employ post-nonlinear models, considering their ease of parameterization and understanding.

### A.9 DISCUSSION ON THE PROPOSED CAUSAL GRAPH

The versatility of our proposed causal graph extends beyond being narrowly designed for domain adaptation. Instead, we hope that it is potential to effectively tackle a wide range of tasks across diverse problem domains. In scenarios such as causal or anti-causal tasks, where a more specific causal graph structure is deemed essential, our proposed causal graph might stand out as a flexible and adaptable framework. It may serve not only as a solution for domain adaptation but also as a source of inspiration for various tasks. This adaptability is rooted in the intentional modeling of latent causal relationships within unstructured observational data. To illustrate, consider its application in segmentation tasks, where interpreting graph nodes as distinct regions. Here, the absence of direct connections between nodes may be reasonable; instead, connections should be contemplated within high-level latent variables. This attribute enhances the versatility and potency of the proposed causal graph, making it adaptable to the nuanced requirements of diverse tasks and problem domains.

### A.10 LIMITATIONS

One of the foundational assumptions mandates significant alterations in the latent noise variables, driving observable variations in the data. However, the pragmatic feasibility of meeting this assumption in real-world applications adds a layer of intricacy. The endeavor to induce substantial changes in latent noise variables encounters challenges rooted in the inherent complexities of data-generating processes. Specifically, in practical applications, the availability of an extensive pool of training data across diverse domains may be limited. This limitation introduces a potential vulnerability for the proposed method. Addressing such scenarios involves effectively leveraging the independence among $n$. For instance, the imposition of independence can be achieved through various regularization techniques, as enforcing a hyperparameter to enhance the independence among $n_i$ in Eq. (7).

