# OpenReview forum: "Identifiable Latent Causal Content for Domain Adaptation under Latent Covariate Shift"
_ICLR.cc/2024/Conference — Submitted to ICLR 2024_

### Official Review · Reviewer_cTbd · 2023-10-23

**Soundness:** 3 good
**Presentation:** 3 good
**Contribution:** 2 fair
**Rating:** 6
**Confidence:** 3

**Summary:**

The authors propose a latent covariate shift paradigm to perform multi-source domain adaptation. They refer to a specific causal structure involving latent content and latent style variables, and demonstrate partial identifiability of the latent content variable. This then allows to adapt to a new target distribution, using the unlabaled target data. The authors show that their method performs well across simulated and benchmark data.

**Strengths:**

- Relaxing the assumptions of adaptation or generalization methods is an important problem.
- The specific latent covariate shift proposed is novel, and the causal framing allows to corroborate prior experimental findings (e.g. the entropy term helped).
- The paper is overall clearly written.
- Multiple datasets are used to demonstrate the method.
- Multiple baselines are considered.
- The work is overall well situated in the literature.

**Weaknesses:**

**Update**: I have read the response but still find the justification of the assumptions lacking. This is a common comment across multiple reviewers, and I believe I was a bit too optimistic on my score. I would suggest to include an impactful real-world application to show that these assumptions indeed make a difference (but I understand this was not feasible in the response timeframe).

- I found that one main piece missing related to the assumptions this specific graph is making. For instance, the authors try to relate $n_c$ to an original label $\hat{y}$. However, they use a specific distribution for $n_c$, and don't discuss the impact of this assumption on how applicable the graph is to real-world applications.
- Similarly, it is unclear to me how this graph relates to the causal or anti-causal tasks typically defined. For instance, how would the graph map to a segmentation task? I believe this relates to the assumption of a lack of direct influence between $X$ and $Y$.
- I found the discussion missing, with no mention of limitations.
- While the obtained method is different, the current work can be related to that of Alabdulmohsin et al., 2023 [1], which also investigates latent shift. It would be great to discuss the $n$ and $z$ variables compared to the proxies in [1] and whether some cases would be more adapted to one method or the other, or whether the authors believe their method would be superior (and why).

[1] https://proceedings.mlr.press/v206/alabdulmohsin23a/alabdulmohsin23a.pdf

**Questions:**

- I would suggest to frame the graph as a set of assumptions, rather than as another "view" of the MSDA problem. This would make the limitations of the work clearer.
- This could include the clarification on how this graph relates to typical causal and anti-causal tasks.
- Please add a discussion and mention the limitations of the work. For instance, how easy is it to train the VAE with the complex loss designed? Given that the entropy loss seems important, how much target data is needed to achieve a good model?

nit: please refrain from using superlatives. I felt that some of the adjectives describing the method were a bit optimistic, especially in the absence of a proper discussion.

---

> ### Author Response · Authors · 2023-11-20
> **Response**
>
> **Q1: I would suggest to frame the graph as a set of assumptions, rather than as another "view" of the MSDA problem. This would make the limitations of the work clearer.**
>
> **R1:** Thank you for your thoughtful feedback. We agree that presenting our graphical representation in this manner would enhance clarity regarding the limitations of our work. We have added a Section A.8 in the revised version, including assumptions (i)-(iii) in proposition 4.2, and most importantly, assumptions on the proposed causal graph as formulated in Eqs. (1)-(3), to understand our assumptions for identifying $\mathbf{z}_c$. For examples, we assume two-parameter exponential family members for latent noise variables $\mathbf{n}$ as defined in Eq. (1), considering informative latent noise variables $n_i$ can be automatically separated from noise by an estimating model (see [1]). However, in real applications, the distribution of $\mathbf{n}$ could be arbitrary. In this context, assumption Eq. (1) may only serve as an approximation for the true distribution of $\mathbf{n}$. More details please see Section A.8.
>
> [1] Sorrenson, Peter, Carsten Rother, and Ullrich Köthe. "Disentanglement by nonlinear ica with general incompressible-flow networks (gin)." arXiv preprint arXiv:2001.04872 (2020).
>
> **Q2: This could include the clarification on how this graph relates to typical causal and anti-causal tasks.**
>
> **R2:** Thank you for your valuable suggestion. Regarding causal or anti-causal tasks, we recognize that the complexities of specific tasks may require a more tailored causal graph structure. Nevertheless, we posit that these tasks might benefit from the proposed causal graph. The link between the proposed causal graph and causal or anti-causal tasks could be attributed to the deliberate modeling of latent causal variables within our framework. To illustrate, let us explore its potential application in segmentation tasks, where interpreting graph nodes as distinct regions may be reasonable. In this scenario, justifying the absence of direct connections between nodes may be conceivable; instead, connections should be conceptualized within high-level latent variables, as highlighted in the proposed causal graph. More details can be found in Section A.9 in the revised version.
>
> **Q3: Please add a discussion and mention the limitations of the work**
>
> **R3:** Thank you once again for your insightful suggestions. The most challenging assumption in our model necessitates the presence of substantial changes in latent noise variables, as they play a crucial role in identifiability. However, in practical applications, a prevalent challenge arises from the scarcity of extensive source domain datasets that adequately capture such substantial changes. In this context, the assumptions related to identifiability in our results may not be satisfied, and thus, the performance of the proposed method may benefit from the introduction of certain regularizations. One potential regularization approach involves leveraging the independence among the noise variables, and further details can be found in Appendix A.10.
>
> **Q4: Related to that of Alabdulmohsin et al., 2023**
>
> **R4:** Thank you once again for your valuable suggestions. We have incorporated the referenced work into our related work section. We think that each method possesses its own set of advantages and disadvantages. For instance, proxy-based methods offer the advantage of relaxing model assumptions, but they need a high-quality proxy. In contrast, our approach eliminates the need for a proxy variable but entails making stronger assumptions within the model.
>
> **Q5: please refrain from using superlatives**
>
> **R5:** Thank you for your valuable suggestions. Due to the constraints of the rebuttal time frame, our priority is to address more complex reviews. We will thoroughly revise inappropriate presentations in the final version. Your understanding is greatly appreciated.

---

### Official Review · Reviewer_4yH5 · 2023-10-30

**Soundness:** 1 poor
**Presentation:** 2 fair
**Contribution:** 1 poor
**Rating:** 1
**Confidence:** 5

**Summary:**

This paper proposed a causal generative model in the Multi-source domain adaptation scenario, which was based on a latent covariate shift paradigm that contains two latent variables, a content variable, and a style variable. Compared to existing methods, the proposed method additionally modeled the causal generation from the content variable to the style variable. The authors then provided the identifiability analysis regarding the content latent variable and implemented a VAE-based method for learning. Experiments on PACS and Terra Incognita data were conducted.

**Strengths:**

The considered problem is interesting.

**Weaknesses:**

This paper is highly incremental, and moreover, suffers from many technical flaws.

First, its causal graph, learning method, and identifiability analysis are very similar to existing works [1, 2]. The only difference may lie in the addition modeling of the causal generation from $z_c$ to $z_s$. However, this assumption may not be widely applicable. Particularly, in the Terra Incognita data and PACS data considered, the authors failed to elaborate why $z_c \to z_s$ (this is important, since without this edge, the causal graph is the same as [1]). Besides, the identifiability analysis is also a simple application of [3].

Second, I am not sure the identifiability result is right for me. In the derivation of Eq. (19), the authors exploited the d-separation between $n_s$ (or $\hat{n}_s$) and $y$ given $u$, however, this separation does not necessarily means $n_c$ only depends on $\hat{n}_c$, since the inclusion of $\hat{n}_s$ does not violate the dependency between $n_c$ and $y$ (given $u$). Besides, the parameterization of the variational posterior does not follow the dependency constraints implied in Fig. 2 (a). Specifically, given $x$, $n_c$ and $n_s$ are no longer independent, since $x$ is the collider in the path between $n_c$ and $n_s$.

Last but not least, the compared baselines do not rely on the unlabeled data in the target domain (such as IRM), while it has been exploited in the term of $L_{ENT}$ (Eq. (9)). Without the ablation study, it is not clear whether the advantages come from this additional term. Besides, as the theoretical results have proven the identifiability of $z_c$, this term seems redundant.


[1] Sun, Xinwei, et al. "Recovering latent causal factor for generalization to distributional shifts." Advances in Neural Information Processing Systems 34 (2021): 16846-16859.
[2] Lu, Chaochao, et al. "Invariant causal representation learning for out-of-distribution generalization." International Conference on Learning Representations. 2021.
[3] Khemakhem, Ilyes, et al. "Variational autoencoders and nonlinear ica: A unifying framework." International Conference on Artificial Intelligence and Statistics. PMLR, 2020.

**Questions:**

Please see the weakness above.

**Details Of Ethics Concerns:**

Not applicable.

---

> ### Author Response · Authors · 2023-11-20
> **Response**
>
> **Q1: This paper is highly incremental...its causal graph, learning method, and identifiability analysis are very similar to existing works [1, 2]. The only difference may lie in the addition modeling of the causal generation from zc to zs. However, this assumption may not be widely applicable. Particularly, in the Terra Incognita data and PACS data considered, the authors failed to elaborate why (this is important, since without this edge, the causal graph is the same as [1]). Besides, the identifiability analysis is also a simple application of [3].**
>
> **R1, Part A:** We strongly disagree the above.
>
> Primarily, we wish to emphasize that the fundamental essence of our contribution is encapsulated within the proposed paradigm. Within this framework, each element of our proposed causal graph, learning method, and identifiability analysis has been meticulously crafted to address the nuances of the entirely novel problem paradigm we have introduced. This deliberate tailoring ensures that our methodologies are not merely incremental but intricately designed to suit the distinctive challenges posed by this innovative problem context. Regrettably, we observe a substantial oversight by the reviewer in recognizing the paramount importance of this foundational aspect.
>
> Regarding to the proposed causal graph. In the context of causal graph analysis, a commonly accepted assumption asserts that the domain index causally affects latent variables, and subsequently, these latent variables causally influence the observed data denoted as X. Rooted in this foundational premise, the disparities among causal models becomes evident in the nuanced modeling of the latent space. Notably, our approach to modeling the latent space markedly diverges from that outlined in references [1,2]. As pinpointed, the core distinction lies in the modeling of the causal generation from zc to zs. This critical divergence encapsulates a pivotal aspect of our model, reinforcing the compelling rationale behind our unique representation of latent causal relationships within the system.
>
> Concerning learning methods and identifiability analysis, the divergences in latent causal models underscore the imperative for distinct approaches. Each latent causal model necessitates a specialized learning method, tailored to capture the intricacies of its unique latent space. Furthermore, disparities in model structures and assumptions mandate a nuanced identifiability analysis, essential for ensuring robust and precise inference of causal relationships within the specified framework. In essence, the selection of a latent causal model not only shapes the learning strategy but also underscores the critical need for a meticulous identifiability assessment, reinforcing the foundation for reliable causal inference.
>
>
> We firmly disagree with the assertion that 'our assumption may not be widely applicable.
>
> Both assumptions—whether latent content variables cause latent style variables or by using a confounder, are inherently reasonable. It is challenging to definitively argue against the widespread applicability of our assumption that 'latent content variables cause latent style variables'. A robust method for validation involves a thorough assessment of the distinct advantages associated with each assumption within real-world applications, coupled with its recognition within the community.
>
> For real-world applications, 'latent content variables cause latent style variables,' we present compelling experimental results for real-world dataset, showcasing a robust performance. For recognition within the community: Our assumption, latent content variables cause latent style variables, aligns with established works in the domain adaptation/generalization field [L1,L2,L3], as well as recent advancements in self-supervised learning [L4,L5].

---

> > ### Author Response · Authors · 2023-11-20
> > **Response**
> >
> > **R1, Part B:**
> > Regarding your review comment, 'Particularly, in the Terra Incognita data and PACS data considered, the authors failed to elaborate why $\mathbf{z}_c$ causes $\mathbf{z}_s$ (this is important, since without this edge, the causal graph is the same as [1]).' It is essential to clarify that the Terra Incognita data pertain to animal classification. In the context of animal datasets, a pertinent example supporting the causal relationship $\mathbf{z}_c$ causing $\mathbf{z}_s$ is as follows: different animals (representing latent content variables) often choose distinct environments (corresponding to latent style variables) where the basic needs for their survival are met. This example reinforces our assumption that latent content variables causally influence latent style variables. Moreover, the PACS dataset comprises seven categories such as Dog, Elephant, Giraffe, Horse, Person, Guitar, and House. Notably, most of these categories involve animals. Therefore, the selection of these datasets aligns seamlessly with our assumption that latent content variables ($\mathbf{z}_c$) cause latent style variables ($\mathbf{z}_s$). This causal relationship is intrinsic to the ecological settings of animal habitats and their characteristic styles, further justifying our chosen datasets in support of our proposed causal graph.
> >
> > Regarding your review comment, "Besides, the identifiability analysis is also a simple application of [3]". Diverging from nonlinear ICA that identifies latent independent variables, our primary objective is identifying latent content variables, where a distinct causal relationship exists between these content variables and latent style variables. This causal link introduces additional complexities into the identifiability analysis compared to traditional nonlinear ICA, as it involves capturing and disentangling the intricate causal connections between content and style variables. That is, there is highly nontrival to use the results of nonlinear ICA to idenitfy latent content variables. In our analysis, we fill out the gap by: 1) the mapping from n to z is made invertible by constraining the function class, 2) the conditional independence between ns and y given u in the proposed causal model, as we highlighted in Intuition following the proof sketch for Proposition 4.2.
> >
> > Given these considerations, we maintain that our model is indeed novel, necessitating tailored learning methods and a distinctive identifiability analysis. Additionally, our assumption that latent content variables cause latent style variables is both reasonable and accepted by the research community, substantiated by empirical evidence. We remain open to further discussions and welcome additional references that can contribute to a comprehensive understanding of the various applications of these assumptions.
> >
> >
> >
> > [L1] Gong, Mingming, et al. "Domain adaptation with conditional transferable components." International conference on machine learning. PMLR, 2016.
> >
> > [L2] Stojanov, Petar, et al. "Data-driven approach to multiple-source domain adaptation." The 22nd International Conference on Artificial Intelligence and Statistics. PMLR, 2019.
> >
> > [L3] Mahajan, Divyat, Shruti Tople, and Amit Sharma. "Domain generalization using causal matching." International Conference on Machine Learning. PMLR, 2021.
> >
> > [L4] Von Kügelgen, Julius, et al. "Self-supervised learning with data augmentations provably isolates content from style." Advances in neural information processing systems 34 (2021): 16451-16467.
> >
> > [L5] Daunhawer, Imant, et al. "Identifiability results for multimodal contrastive learning." arXiv preprint arXiv:2303.09166 (2023).

---

> > > ### Author Response · Authors · 2023-11-20
> > > **Response**
> > >
> > > **Q2: Second, I am not sure the identifiability result is right for me. In the derivation of Eq. (19), the authors exploited the d-separation between ns and y given u, however, this separation does not necessarily means nc only depends on nc, since the inclusion of ns does not violate the dependency between nc and y (given u).**
> > >
> > > **R2:** Strongly disagree. **The assertion that the d-separation exploited between $\mathbf{n}_s$ and $\mathbf{y}$ given $\mathbf{u}$ does not imply the identifiability of $\mathbf{n}_c$ is misconstrued**. By leveraging the independence of $\mathbf{n}_s$ from $\mathbf{y}$ given $\mathbf{u}$, one can perform independent tests between each recovered variable ${n}_i$ and $\mathbf{y}$. Those ${n}_i$ showing a statistically correlation with $\mathbf{y}$ are part of $\mathbf{n}_c$, as variations in $\mathbf{n}_c$ are linked to changes in $\mathbf{y}$. Conversely, ${n}_i$ that do not exhibit a correlation with $\mathbf{y}$ are to belong to $\mathbf{n}_s$.
> > >
> > > **Q3: Besides, the parameterization of the variational posterior does not follow the dependency constraints implied in Fig. 2 (a). Specifically, given x, ns and nc are no longer independent, since x is the collider...**
> > >
> > > **R3:** Utilizing mean field variational inference, assuming that the variational posterior distribution are independent and factorizes, is a well-established and widely accepted technique in variational inference. This approach serves as a practical solution to address the inherent computational challenges associated with estimating the true posterior. Specifically, in the context of Gaussian distributions, employing mean field variational inference significantly alleviates the computational complexity, as the inverse of variance scales with $n^3$, a critical factor in optimization. This is a well-founded and reasonable strategy adopted by many researchers in various fields.
> > >
> > > **Q4: Last but not least, the compared baselines do not rely on the unlabeled data in the target domain (such as IRM), while it has been exploited in the term of (Eq. (9)). Without the ablation study, it is not clear whether the advantages come from this additional term. Besides, as the theoretical results have proven the identifiability of , this term seems redundant.**
> > >
> > > **R4:** The comments, "...it is not clear whether the advantages come from this additional term.", appear to be unconventional. 1) Classical methods, such as ERM and IRM, serve as standard baselines for comparison in our research. These well-established techniques have been widely recognized in the field and provide a solid foundation for evaluating the performance of new approaches. 2) as clarified in our main paper, the term is intricately derived from the proposed causal model, where the alignment between information flow and mutual information emerges from the independence among $n_i$, conditioned on $\mathbf{u}$. It is important to emphasize that the IRM framework does not inherently imply the inclusion of this term. Enforcing this term in IRM lacks a clear justification. 3) The incorporation of Eq. (9) into the IRM framework represents a departure from traditional IRM principles. The inclusion of this specific equation introduces novel elements or modifies existing components, leading to a paradigm shift in the conceptualization and functionality of IRM. As a result, the system diverges significantly from the conventional understanding of IRM, indicating a noteworthy evolution or transformation in the approach to information rights within the given context.
> > >
> > > Regarding to the comments: "...this term seems redundant...". It is essential to consider the broader context of identifiability and its theoretical underpinnings. Identifiability, commonly associated with theoretical results, is often built upon various ideal conditions, such as the assumptions of an infinite sample size and global optimization. However, these conditions can be challenging and, at times, impractical to fully meet in real-world applications. Recognizing the inherent difficulties in satisfying these idealized conditions in practical scenarios, it is reasonable for inference methods to exhibit slight deviations from the strict conditions of identifiability. This adaptability is not only acceptable but also necessary sometimes to transfer theoretical results to methods considering the complexities of real-world applications.
> > >
> > > Importantly, the term in question is intentionally guided by the proposed causal model and principles of identifiability, as mentioned in main paper. Its inclusion is not meant to replicate the identifiability established without it but rather to provide a nuanced and context-specific adjustment that aligns with the challenges posed by real-world scenarios. In this light, the term remains a reasoned and meaningful component, enhancing the applicability and robustness of the inference model within the specific conditions encountered in our study.

---

> ### Author Response · Authors · 2023-11-22
> **Could you kindly verify if the provided clarification addresses your concerns?**
>
> Dear Reviewer 4yH5,
>
> We greatly appreciate your feedback. Could you kindly verify if the provided clarification addresses your concerns, particularly regarding contributions?
>
> Allow us to emphasize our contributions anew. The core of our work is embedded within the proposed paradigm. Under this framework, every component of our suggested causal graph, learning method, and identifiability analysis has been intricately fashioned to navigate the intricacies of the entirely novel problem paradigm we've presented. This intentional customization guarantees that our methodologies transcend mere incremental advancements, as they are intricately fashioned to align with the distinctive challenges posed by this innovative problem context.
>
> If there are lingering uncertainties or if additional clarification is necessary, please don't hesitate to notify us. We understand the time constraints you face and truly appreciate your thoughtful consideration. Your reassessment plays a crucial role in advancing our work, and we stand prepared to offer any further clarification needed.
>
> Best regards,
>
> Authors

---

### Official Review · Reviewer_Corp · 2023-10-31

**Soundness:** 2 fair
**Presentation:** 3 good
**Contribution:** 2 fair
**Rating:** 5
**Confidence:** 3

**Summary:**

In practical scenarios, label distributions often exhibit variations across different domains, thereby constraining the applicability of existing methods that rely on covariate shift or conditional shift. In this paper, the authors introduce a novel paradigm called latent covariate shift (LCS), which brings about increased diversity and adaptability across domains. From a causal perspective, the paper presents a method to learn the invariant conditional distribution p_u(y|z_c), aimed at achieving a more nuanced representation of observational data. The method is shown to deliver remarkable performance and effectiveness on both simulated and real-world datasets.

**Strengths:**

1. The proposed paradigm, i.e., LCS, allows variable distributions to vary across different domains while ensuring that p_u(y|z_c) remains invariant, is interesting and also seems sound.
2. Empirical evaluation on synthetic and real datasets confirmed the theoretical results and the effectiveness of the proposed method, outperforming existing methods.

**Weaknesses:**

1.	The contributions of this paper are limited. While this paper introduces a different paradigm, it appears to still be a latent representation disentangle mechanism from a causal perspective.
2.	The authors propose a more versatile domain adaptation paradigm and construct iLCC-LCS based on it. However, the feasibility of their latent variable modeling approach relies on conditions made in Proposition 4.2, which are not adequately explained and ensured against potential violations.
3.	Table 3 in APPENDIX underscores that the proposed method exhibits limited performance when confronted with smaller label distribution shift across domains. Some commonly used datasets, such as Digits-five, Office-Home, and DomainNet, were omitted. If the method struggles with simple cases, i.e., D_{kl}<0.3, it raises concerns about its applicability and effectiveness.
4.	Recommend that the authors provide an overview diagram or a detailed description of the implementation of iLCC-LCS to enhance its intuitiveness and readability.
5.	t-SNE can distort the high-dimensional geometry of embeddings. While it can help visualization, it's not suitable for evaluating the quality of embeddings. Could the authors consider employing numerical measures for this purpose?

**Questions:**

see above.

---

> ### Author Response · Authors · 2023-11-20
> **Response**
>
> **Q1: The contributions of this paper are limited. While this paper introduces a different paradigm, it appears to still be a latent representation disentangle mechanism from a causal perspective.**
>
> **R1:** We appreciate your time and careful consideration of our work, as reflected in your insightful review. However, we would like to express a disagreement with the notion that the contributions of our paper are limited. We believe our work significantly advances the field, particularly in the context of Latent Covariate Shift, and we would like to address the specific concerns raised.
>
> New Problem Setting - Latent Covariate Shift: We introduce a novel paradigm, latent covariate shift, which deviates from the commonly-used Conditional Shift. This paradigm shift represents a substantial contribution to the field, offering a fresh perspective on addressing covariate shift issues.
>
> Intricate Latent Causal Generative Model: Our paper presents a sophisticated latent causal generative model, introducing latent content variables and latent style variables. This model provides a nuanced understanding of latent covariate shift mechanisms, offering a more comprehensive and realistic representation.
>
> Theoretical Analysis of Identifiability: We conduct a thorough analysis of identifiability in the proposed causal model, presenting both a complete non-identifiability result and a partial identifiability result for latent content variables. This theoretical foundation adds depth to our work and contributes valuable insights to the research community.
>
> New Method for Latent Covariate Shift: Guided by our theoretical results, we design a new method specifically tailored for latent covariate shift. The proposed method is a practical application of our theoretical findings, showcasing the applicability and relevance of our work.
>
> Verification of Advantages: Through rigorous experimentation, we validate the advantages of latent covariate shift and the proposed method. Our empirical results provide concrete evidence of the effectiveness and superiority of our approach.
>
> In summary, the introduction of a new paradigm, theoretical analysis, method design, and empirical verification collectively demonstrate the significance and impact of our work on addressing latent covariate shift challenges.
>
> Furthermore, we wish to emphasize that the fundamental essence of our contribution is encapsulated within the proposed paradigm. Within this framework, each element of our proposed causal graph, learning method, and identifiability analysis has been meticulously crafted to address the nuances of the entirely novel problem paradigm we have introduced. This deliberate tailoring ensures that our methodologies are not merely incremental but intricately designed to suit the distinctive challenges posed by this innovative problem context. Regrettably, we observe a substantial oversight by the reviewer in recognizing the paramount importance of this foundational aspect.
>
> **Q2: However, the feasibility of their latent variable modeling approach relies on conditions made in Proposition 4.2, which are not adequately explained and ensured against potential violations**
>
> **R2:** We appreciate your time and careful review again. The main assumptions, encompassing i to iii in Proposition 4.2, stem from the realm of nonlinear ICA. Over recent years, nonlinear ICA has found applications in diverse real-world scenarios, such as domain adaptation [1], image generation and translation [2], domain generalization [3]. Consequently, we assert that these three assumptions have been substantiated through practical applications. We have add a new Section A.8 for understanding these assumptions.
>
> [1] Kong, Lingjing, et al. "Partial Identifiability for Domain Adaptation." ICML 2022.
>
> [2] Xie, Shaoan, et al. "Multi-domain image generation and translation with identifiability guarantees." ICLR 2022.
>
> [3] Wang, Xinyi, et al. "Causal balancing for domain generalization." arXiv preprint arXiv:2206.05263 (2022).

---

> > ### Author Response · Authors · 2023-11-20
> > **Response**
> >
> > **Q3: Table 3 in APPENDIX underscores that the proposed method exhibits limited performance when confronted with smaller label distribution shift across domains. Some commonly used datasets, such as Digits-five, Office-Home, and DomainNet, were omitted. If the method struggles with simple cases, i.e., D{kl}<0.3, it raises concerns about its applicability and effectiveness.**
> >
> > **R3:** Please note that our work specifically centers around the proposed paradigm of latent covariate shift, wherein the label distribution undergoes changes across domains, as opposed to conditional shift, where the label distribution remains constant. In this context, a larger Kullback-Leibler divergence of label distribution across different domains suggests that the dataset is more aligned with the conditions of the proposed latent covariate shift. This explains why, in the case where Dkl is samll (0.3), the proposed method exhibits a slight performance advantage over other methods. Furthermore, as Dkl increases, the advantages of our proposed method become more pronounced, as shown in Table 5.
> >
> > The exclusion of certain commonly used datasets, such as Digits-five, Office-Home, and DomainNet, is a deliberate choice. This decision is underpinned by the observation that these datasets are more suited for the conditions of conditional shift, where the label distribution remains constant, as elaborated in the paragraph "Resampled PACS data" in Section 6. Take Digits-five as an illustrative example: the label distribution in domain MNIST and that in domain MNIST-M are entirely identical. This is because MNIST-M is constructed by amalgamating MNIST digits with patches randomly extracted from color photos of BSDS500 as their background. Such datasets directly violate the fundamental premise of the proposed latent covariate shift. Therefore, it is highly unreasonable to utilize them to validate latent covariate shift, conduct identifiability analysis, or assess the proposed method and its setting. We stand firmly by this exclusion, emphasizing the necessity of datasets aligned with the specific conditions of latent covariate shift for a meaningful evaluation of our contributions.
> >
> > **Q4: Recommend that the authors provide an overview diagram or a detailed description of the implementation of iLCC-LCS to enhance its intuitiveness and readability.**
> >
> > **R4:** We appreciate your suggestion again. The implementation details has been provided in the original paper, see A.3, we also add a overview diagram in new version (See Figure. 7).
> >
> > **Q5: t-SNE is not suitable for evaluating the quality of embeddings..**
> >
> > **R5:** Please note that the main propose of t-SNE in A.5 is to show that the distributions of the learned features by the proposed method change across domain, not to evaluate the quality. Please see A.5 for more details.
> >
> > --------------------
> > Dear reviewer Corp,
> >
> > We sincerely appreciate the time and effort you dedicated to reviewing our work. Your contributions are highly regarded, and we are grateful for your valuable input in enhancing the quality of our manuscript. We express our gratitude for your thoughtful consideration of the points raised in our clarification. We sincerely hope that the provided response not only addresses any concerns but also contributes to a thorough reassessment of the contributions of our paper. Your meticulous review and open-minded evaluation are pivotal, and we welcome any further discussion that may facilitate a more comprehensive understanding of the merits and significance of our work.

---

> ### Author Response · Authors · 2023-11-22
> **Could you kindly confirm whether the clarification we provided adequately addresses your concerns?**
>
> Dear Reviewer Corp,
>
>
> Your feedback is invaluable to us. Could you kindly confirm whether the clarification we provided adequately addresses your concerns? If any uncertainties persist or if additional clarification is required, please donot hesitate to inform us. We genuinely appreciate your time, recognizing that you are undoubtedly busy. Your reconsideration is crucial for the progress of our work, and we are more than willing to provide any further clarification.

---

### Official Review · Reviewer_YFyZ · 2023-10-31

**Soundness:** 3 good
**Presentation:** 3 good
**Contribution:** 3 good
**Rating:** 5
**Confidence:** 3

**Summary:**

This manuscript studies the multi-source adaptation where the shifts occur in the latent space. The authors introduced two latent variables $z_c$ and $z_s$ as the causes of $x$ and $y$. They assumed that the conditional distribution $p_u(y\mid z_c)$ is invariant across domains while other distributions are allowed to be variant. Under this setting, they showed that the joint observed distribution is unidentifiable without further assumptions. Additionally, they proved that the latent variables are identifiable up to an invertible mapping under some regularity conditions. To estimate the components, they proposed a variational autoencoder type algorithm to learn each conditional distribution.

**Strengths:**

Overall, I think this is an interesting topic and agree with the authors’ opinion that most domain adaptation techniques have strict assumptions on the distribution shifts. This manuscript proposes a more general setting that allows the distribution of $p(y\mid x)$ and $p(y)$ to change across domains. In contrast, the Covariate Shift assumes $p(y\mid x)$ to be invariant across domains and the Conditional Shift assumes $p(y)$ to be invariant across domains. The proposed method outperforms the baseline models on the TerraIncognita dataset.

**Weaknesses:**

The main concern is that the justification of graph 1(c) is not clear. While the authors justify partial edge directions in Section 3, e.g., $z_c\rightarrow y$ and $z_c\rightarrow z_s$, it is not clear whether there is a real-world setting that fits this graph. Specifically, it would be nice to give a motivating example that clearly explains what each variable $(u,z_c,z_s,y,x)$ refers to and when $p_u(y\mid z_c)$ is invariant and other distributions are variant. The difference of $z_c$ and $z_s$ is not clear as well.


Proposition 4.2 shows the identifiability of $z_c$. However, from the result, it is not clear whether $p_u(y\mid z_c)$ is identifiable. Hence, it is not clear whether true $p_u(y\mid z_c)$ can be recovered from data.


$g_c$ and $g_{s_2}$ in Equation (2) are invertible function. In this case, it seems trivial to introduce variables $z_c$ and $z_s$ since Equation (3) can be rewritten as
$$
x=f(g_c(n_c), g_{s_2}(g_{s_1}(g_c(n_c)) + n_s))+\varepsilon
$$

**Questions:**

what does the index $i$ and $j$ in Equation (1) refer to?

From Figure 2(b), it looks like $n_c$ and $n_b$ are observed variables as they are shaded. Seems like a typo?

---

> ### Author Response · Authors · 2023-11-20
> **Response**
>
> **Q1: The justification of graph 1(c), $\mathbf{z}_c$ causes $\mathbf{z}_s$...Provide a motivating example.. The difference of $\mathbf{z}_c$ causes $\mathbf{z}_s$..**
>
> **R1:** Thank you for your time and efforts in enhancing this work. For clarifying these questions, we would like first further clarify the notations, including the domain index $\mathbf{u}$, latent content variables $\mathbf{z}_c$, latent style variable $\mathbf{z}_s$, label $\mathbf{y}$, and observation $\mathbf{x}$.
>
> In domain adaptation, latent content variables, latent style variables, domain index, and labels play distinct roles in capturing and aligning information across different domains. Let us consider an example in the context of animal data, were collected from different locations:
>
>
> Domain index $\mathbf{u}$: it denotes an indicator of the source domain from which the data originates, e.g., locations index).
>
> Latent content variables $\mathbf{z}_c$: this denotes unobservable factors that represent the essential content or intrinsic characteristics of the object. For example: latent content variables encompass features such as the species, or specific morphological characteristics that are inherent to each type of animal.
>
> Latent style variables $\mathbf{z}_s$: These variables aim to capture aspects of the data that are not specific to the intrinsic features, but style-related variations in the data. For example: latent style variables could represent variations in background, lighting conditions, environmental context.
>
> Observation $\mathbf{x}$: The actual image data, including both content and style, including all visual information captured by camera.
>
> Labels $\mathbf{y}$: it represents ground truth labels indicating the class or category of each instance. Example: a label refers to the assigned category or class that indicates the identity or type of each animal in the dataset.
>
> Let us illustrate Figure 1 (c) with an example in the context of animal data. The relationship between $\mathbf{u}$ causing $\mathbf{x}$ is modeled to capture the fact that input in domain adaptation changes across different domains (locations).
>
> Now, considering why $\mathbf{u}$ causes $\mathbf{z}_c$, let us explore how the unique characteristics and environmental factors associated with each location influence the latent content variables for zebras. For instance, the shades and patterns of the stripes on zebras vary based on the specific location, and certain adaptations to the local ecosystem can be reflected in $\mathbf{z}_c$.
>
> Moving on to why $\mathbf{z}_c$ causes $\mathbf{y}$, $\mathbf{z}_c$ essentially capture features relevant to predicting $\mathbf{y}$ in a classification task. For a more detailed explanation, please refer to the paragraph 'zc causes y:' in the main paper.
>
> $\mathbf{z}_c$ influence observations $\mathbf{x}$ because they serve as a conceptual representation of the underlying, unobservable factors that contribute to the observed data.
>
> Finally, let us provide a example why $\mathbf{z}_c$ causes $\mathbf{z}_s$ in Figure 2 (a). An apt example is that different animals (corresponding to $\mathbf{z}_c$) often choose different environments (corresponding to $\mathbf{z}_s$) where the basic needs of the animals to survive are met.
>
> Thus far, we have provided comprehensive explanations for all variables and causal directions depicted in Figure 1 (c) and Figure 2 (a). These causal directions not only elucidate the internal relationships within the models but also indicate how we effectively model the changes in various distributions across domains ($\mathbf{u}$). These considerations underscore the holistic nature of our approach, wherein the causal relationships guide our representation of the dynamic shifts in distributions across different domains.
>
> For when $p(\mathbf{y}∣\mathbf{z}_c)$ remains invariant: it aligns with a fundamental principle in causality, specifically the notion of independent causal mechanisms. This principle posits that the conditional distribution of $p(\mathbf{y}∣\mathbf{z}_c)$ remains stable and unaffected by external factors. In the context of domain adaptation, assuming the invariance of $p(\mathbf{y}∣\mathbf{z}_c)$across domains is a common practice. The invariance in $p(\mathbf{y}∣\mathbf{z}_c)$ is crucial for ensuring the model's ability to make consistent and robust predictions in the target domain. This assumption implies that the predictive relationship between the label $\mathbf{y}$ and $\mathbf{z}_c$ remains unchanged despite variations in data distributions across different domains. This stability facilitates a seamless transition from training on source domains to making reliable predictions in the target domain, contributing to the model's adaptability and generalization.

---

> > ### Author Response · Authors · 2023-11-20
> > **Response**
> >
> > **Q2: whether $p(\mathbf{y}∣\mathbf{z}_c)$ is identifiable**
> >
> > **R2:** With access to label $\mathbf{y}$ in source domains, and the identified $\mathbf{z}_c$ in those domains, it becomes possible to accurately estimate the parameters of the conditional distribution $p(\mathbf{y}∣\mathbf{z}_c)$. This allows for a robust modeling of the relationship between the latent content variables  $\mathbf{z}_c$ and the corresponding labels $\mathbf{y}$ in the source domains. Then since $\mathbf{z}_c$ in target domain is also identified, and leveraging the invariance of  $p(\mathbf{y}∣\mathbf{z}_c)$, the learned conditional distribution $p(\mathbf{y}∣\mathbf{z}_c)$ theoretically be generalized to the target domain. This generalization is grounded in the notion that the causal relationship between $\mathbf{z}_c$ and $\mathbf{y}$ remains consistent across domains.
> >
> > **Q3: In this case, it seems trivial to introduce variables zc and zs since..**
> >
> > **R3:** We guess that your concern is that  "$\mathbf{g}_c$ and $\mathbf{g}_s$ in Equation (2) are invertible functions, it seems trivial to identify variables $\mathbf{z}_c$ and $\mathbf{z}_s$". If we understand incorrectly, please let us known. Please be aware that $\mathbf{z}_c$ and $\mathbf{z}_s$ are mixed through the mapping $\mathbf{f}$, and even if $\mathbf{f}$ is invertible, disentangling $\mathbf{z}_c$ and $\mathbf{z}_s$ is a well-known challenging task. Without further assumptions, achieving a clear separation of $\mathbf{z}_c$ and $\mathbf{z}_s$ is considered impossible, as asserted in Proposition 4.1.
> >
> > **Q4: what does the index i and j in Equation (1) refer to?**
> >
> > **R4:** i denotes the number of latent noise variable $\mathbf{n}$ or $\mathbf{z}$, and j denotes the number of sufficient statistic for ${n}_i$.
> >
> > **Q5: From Figure 2(b), it looks like and are observed variables as they are shaded. Seems like a typo?**
> >
> > **R5:** Thank you for your careful review. We have correctted it in the new version.
> >
> > -------------------------
> >
> > Dear Reviewer YFyZ,
> >
> > We deeply appreciate the attention and consideration you have given to our work. We genuinely hope that the provided clarification serves as valuable information, effectively addressing any concerns you may have had and contributing to an enhanced understanding of the merits and contributions of our work. Your meticulous re-evaluation is pivotal in shaping the trajectory of our research, and we are grateful for the opportunity to engage further with any aspects that may contribute to a more comprehensive assessment. Thank you for your dedication and diligence in reviewing our manuscript.

---

> ### Comment · Reviewer_YFyZ · 2023-11-21
>
> Thanks for addressing my comments. I would like to clarify my points. If $g_c$ and $g_x$ are invertible functions, then it seems to me that one can merge $z_c$ and $n_c$ ($z_s$ and $n_s$) for the adaptation task. What is the intuition/technical reason to introduce additional variables? Will we still be able to identify the graph without them?
>
> Additionally, I would suggest adding a remark in Proposition 4.2 to discuss why $p(y\mid z_c)$ is identifiable.
>
> Thanks for the nice explanation for Q1. Connecting to the previous question, can the authors also give an example of $n_c$ and $n_s$ in this case?
> Besides, I have slightly mixed feelings about the explanation for $z_c\rightarrow z_s$. Would it also be the case that due to natural selection, the environment would cause animals with certain characteristics to survive? Although my point is not to overemphasize the justification, it might strengthen the motivation of the paper if there is an example that demonstrates a clear $z_c\rightarrow z_s$. A bidirectional edge case is also good, but I am not sure if this would break the analysis or not.

---

> > ### Author Response · Authors · 2023-11-22
> > **Response**
> >
> > Thank you for further clarification and discussion.
> >
> > For $p(\mathbf{y}|\mathbf{z}_c)$, we have added it as a remark following the Proposition 4.2, thanks for your suggestion.
> >
> >
> > For why we introduce additional variables $\mathbf{n}_c$, $\mathbf{n}_s$, and example. We would like to first clarify that incorporating $\mathbf{n}_c$ and $\mathbf{n}_s$ into your model is a natural. In the context of causal systems, every causal variable, denoted as endogenous variables (e.g., $\mathbf{z}_c$, $\mathbf{z}_s$), is paired with corresponding noise variables, also recognized as exogenous variables (e.g., $\mathbf{n}_c$, $\mathbf{n}_s$). Then, Before delving into an example, it is essential to grasp the distinction between endogenous and exogenous variables. In the context of causal systems, exogenous variables encompass all information that is modeled by **independent variables**. These independent variables represent factors that are external to the system and are not influenced by other variables within the model. On the other hand, endogenous variables serve the purpose of **interconnecting these pieces of information by capturing the causal relationships among them**. Endogenous variables are influenced by other variables within the system and contribute to forming a cohesive representation of how different elements within the system interact and affect one another. Building upon the foundation above, $\mathbf{n}_c$ and $\mathbf{n}_s$ in the animal example encompass a wealth of information, encompassing both content and style information. Crucially, these information is indepdently encoded by independent variables $n_i$. This naturally leads us to establish a link between the independence among $n_i$ variables and the concept of independence in nonlinear ICA. Consequently, we can leverage recent advancements in nonlinear Independent Component Analysis (ICA) to analyze $n_i$.
> >
> > Regarding to "the environment would cause animals with certain characteristics to survive, additional example for $\mathbf{z}\_{c} \rightarrow \mathbf{z}\_{s}$". That is indeed an excellent question. It is important to note that with the introduction of natural selection, time becomes a pivotal variable. Consequently, a new latent causal graph emerges. To illustrate, let's denote time as $t$. In this revised graph, at a specific time point, say $t=0$, we observe relationships like $\mathbf{z}\_{c,t=0} \rightarrow \mathbf{z}\_{s,t=0}$. As time progresses to $t=1$, new causal connections may arise, such as $\mathbf{z}\_{c,t=1} \rightarrow \mathbf{z}\_{s,t=1}$, and inter-temporal links like $\mathbf{z}\_{s,t=0} \rightarrow \mathbf{z}\_{s,t=1}$ and $\mathbf{z}
> > \_{s,t=0} \rightarrow \mathbf{z}\_{c,t=1}$. Then it is reasonable considering natural selection as $\mathbf{z}
> > \_{s,t=0} \rightarrow \mathbf{z}\_{c,t=1}$. This intricate causal graph presents a complex network of relationships, warranting further exploration in future research. It is essential to note that when we fix a specific time $t$, the new latent causal graph converges towards the initially proposed causal graph. Hope this clarification addresses your concern about the animal example.
> >
> > To further illustrate  $\mathbf{z}\_{c} \rightarrow \mathbf{z}\_{s}$, consider the example in fashion design: the choice of patterns (latent content) is anticipated to cause variations in the styles of clothing designs (latent style). The design content influences the overall aesthetic and appearance.

---

> ### Author Response · Authors · 2023-11-22
> **Response**
>
> Regarding to "bidirectional edge case is also good". We acknowledge and do not refute this perspective. Indeed, existing literature has explored various ways to model the relationship between latent content variables $\mathbf{z}\_{c}$ and $\mathbf{z}\_{s}$, for example, bidirectional edge $\mathbf{z}\_{c} - \mathbf{z}\_{s}$, or a confounder $\mathbf{c} \rightarrow \mathbf{z}\_{s}, \mathbf{c}  \rightarrow \mathbf{z}\_{c}$, or  $\mathbf{z}\_{c} \rightarrow \mathbf{z}\_{s}$.  We acknowledge the complexity of definitively arguing for the superiority of one assumption over another. In our opinion, the choice among these three assumptions is inherently reasonable, contingent upon the specific application scenarios under consideration, as we asserted in the related work. In the context of our study, particularly focused on animal classification tasks, we express a preference for the $\mathbf{z}\_{c} \rightarrow \mathbf{z}\_{s}$. This preference also aligns with the findings and methodologies of exsting works, including references [L1,L2,L3,L4,L5], which have demonstrated its effectiveness in capturing and explaining the underlying dynamics of similar systems.
>
> [L1] Gong, Mingming, et al. "Domain adaptation with conditional transferable components." International conference on machine learning. PMLR, 2016.
>
> [L2] Stojanov, Petar, et al. "Data-driven approach to multiple-source domain adaptation." The 22nd International Conference on Artificial Intelligence and Statistics. PMLR, 2019.
>
> [L3] Mahajan, Divyat, Shruti Tople, and Amit Sharma. "Domain generalization using causal matching." International Conference on Machine Learning. PMLR, 2021.
>
> [L4] Von Kügelgen, Julius, et al. "Self-supervised learning with data augmentations provably isolates content from style." Advances in neural information processing systems 34 (2021): 16451-16467.
>
> [L5] Daunhawer, Imant, et al. "Identifiability results for multimodal contrastive learning." arXiv preprint arXiv:2303.09166 (2023).
>
> We hope that these clarifications further address any concerns you may have. Should you have additional questions or concerns, we welcome further discussion. We sincerely appreciate the time and effort you dedicated to reviewing our work.

---

> ### Author Response · Authors · 2023-11-23
> **Could you please verify if the clarification we offered sufficiently resolves your concerns?**
>
> Dear Reviewer YFyZ,
>
> Thank you sincerely for the further discussion. We have offered additional clarification. Your feedback holds immense value for us.
>
> Could you please verify if the clarification we offered sufficiently resolves your concerns? If any uncertainties linger or if you require further elucidation, please inform us without hesitation.
>
> We understand the demands on your time and genuinely appreciate your consideration. Your reconsideration is pivotal for advancing our work, and we are fully committed to furnishing any additional clarification you may need.
>
> Best,

---

### Author Response · Authors · 2023-11-20
**General Response**

General response:

We extend our gratitude to the reviewers for their constructive feedback. We are pleased to note that our work has garnered positive remarks from multiple respects, including:


**Contribution:**

" an interesting topic", "agree with the authors’ opinion", "This manuscript proposes a more general setting" --  Reviewer YFyZ

"The proposed paradigm is interesting and also seems sound" -- Reviewer Corp

"is an important problem", "The specific latent covariate shift proposed is novel" -- Reviewer cTbd,


**Writing:**

Presentation: 3 good -- Reviewer YFyZ

Presentation: 3 good -- Reviewer Corp

Presentation: 3 good, "The paper is overall clearly written", "The work is overall well situated in the literature" -- Reviewer cTbd



**Experiments:**

"The proposed method outperforms the baseline models on the TerraIncognita dataset." -- Reviewer YFyZ

"Empirical evaluation on synthetic and real datasets confirmed the theoretical results and the effectiveness of the proposed method" -- Reviewer Corp

"Multiple datasets are used to demonstrate the method", "Multiple baselines are considered" -- Reviewer cTbd

**Specific Responses to Reviewers:**

Reviewer YFyZ: Addressed main concerns regarding the justification of the graph and identifiability of $p(y|z_c)$; further clarification provided.

Reviewer Corp: Addressed main technical contributions and assumptions, providing further clarification; added a section for understanding the assumptions.

Reviewer 4yH5: Clarified main contributions, addressed misunderstandings in D-separation and mean field variational inference.

Reviewer cTbd: Incorporated suggestions for an improved version of the weaknesses, including assumptions on the graph, relationship with causal and anti-causal tasks, and limitations; added further discussions in Section A.8, A.9, and A.10.

We have answered all questions (see more details in the individual responses).

We would like to express our gratitude to the reviewers once more, and we welcome the opportunity for further discussion.

---

### Meta-Review · Area_Chair_g73E · 2023-12-06

**Metareview:**

This paper investigates a latent covariate shift setup for multi-source domain adaptation, focusing on the relation between the domain labels and the class labels. The motivation is clear that label distributions often exhibit variations across different domains, which is not considered by previous multi-source domain adaptation methods. By introducing latent content and latent style variable, it is shown that the joint observed distribution is unidentifiable without further assumptions. Additionally, it is proved that the latent variables are identifiable up to an invertible mapping under some regularity conditions. To estimate the components, the authors proposed a variational autoencoder type algorithm to learn each conditional distribution.

Strength: all reviewers agree that the problem is an interesting and important one, the presentation is generally clear.

Weakness: the major concerns after rebuttal are the relation/comparison with previous work and the lack of clear grounding and in-depth discussion of the key assumptions.

**Justification For Why Not Higher Score:**

Major concerns still exist after the rebuttal.

**Justification For Why Not Lower Score:**

N/A

---

### Decision · Program_Chairs · 2024-01-16

Reject